# CoCoReviewBench: A Completeness- and Correctness-Oriented Benchmark for AI Reviewers

Hexuan Deng [1 2 *]   Xiaopeng Ke [1 *]   Yichen Li [2 3 *]   Ruina Hu [2 4 *]   Dehao Huang [2]
Derek F. Wong [5]   Yue Wang [2 +]   Xuebo Liu [1 +]   Min Zhang [1]

## Abstract

Despite the rapid development of AI reviewers, evaluating such systems remains challenging: metrics favor overlap with human reviews over correctness. However, since human reviews often cover only a subset of salient issues and sometimes contain mistakes, they are unreliable as gold references. To address this, we build category-specific benchmark subsets and skip evaluation when the corresponding human reviews are missing to strengthen *Co*mpleteness. We also leverage reviewer–author–meta-review discussions as expert annotations and filter unreliable reviews accordingly to strengthen *Co*rrectness. Finally, we introduce CoCoReviewBench, which curates 3,900 papers from ICLR and NeurIPS to enable reliable and fine-grained evaluation of AI reviewers. Analysis shows that AI reviewers remain limited in correctness and are prone to hallucinations, and highlights reasoning models as more effective reviewers, motivating further directions for improving AI reviewers. Benchmarks and models are available at https://github.com/hexuandeng/CoCoReviewBench.

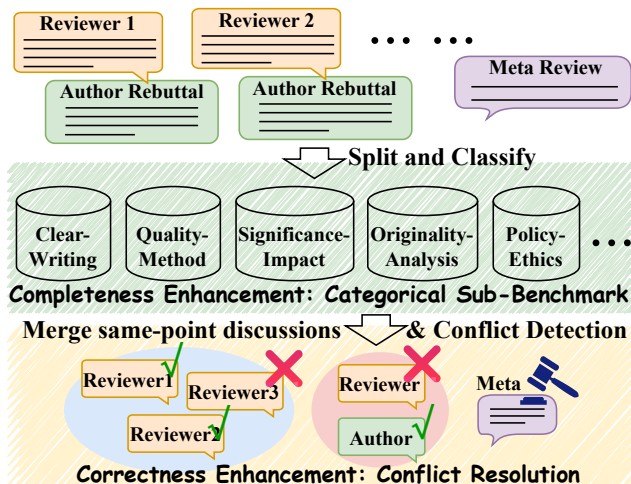

*Figure 1.* Overview of CoCoReviewBench. To tackle incompleteness, we build per-category subsets which contain only papers with reviews in that category. To address incorrectness, we merge discussions about the same point across the reviewers and authors. When explicit conflicts exist, we use the meta-review as a high-level adjudication signal for the final decision.

## 1. Introduction

Peer review is a cornerstone of scientific progress (Hillard & Baber, 2021). However, with the rapid surge in submission volume, reviewers are bearing increasing pressure, leading to an inevitable decline in review quality (Kim et al., 2025; Beecher & Wang, 2025; Morley et al., 2025). To address this, Large Language Models (LLMs) hold significant potential to become powerful assistants that can alleviate these problems (Mann et al., 2025; Thakkar et al., 2025), and also spur substantial efforts to build AI reviewer systems (Liu & Shah, 2023; Jin et al., 2024; Weng et al., 2025).

Therefore, evaluating AI reviewers becomes crucial. A line of work directly adopts the LLM-as-a-judge (Zheng et al., 2023) paradigm without using human opinions as references (Xu et al., 2025; Zhang & Abernethy, 2025). However, LLMs may lack sufficient domain expertise, and can exhibit biases that inflate scores without improvement (Dorner et al., 2025; Krumdick et al., 2025). Meanwhile, more works directly use human reviews as gold references and evaluate similarity to the reference rather than correctness via rule-based matching or LLM-as-a-judge, thereby avoiding the need for expert adjudication and mitigating bias (D'Arcy et al., 2024; Chang et al., 2025; Shin et al., 2025).

[1]Institute of Computing and Intelligence, Harbin Institute of Technology, Shenzhen, China [2]Zhongguancun Academy, Beijing, China [3]Department of Computer Science and Engineering, Southern University of Science and Technology, Shenzhen, China [4]Faculty of Computing, Harbin Institute of Technology, Harbin, China [5]NLP²CT Lab, Department of Computer and Information Science, University of Macau, China. Correspondence to: Yue Wang <yuewang@bza.edu.cn>, Xuebo Liu <liuxuebo@hit.edu.cn>.

*Proceedings of the 43rd International Conference on Machine Learning*, Seoul, South Korea. PMLR 306, 2026. Copyright 2026 by the author(s).

However, the nature of peer review makes it unsuitable as a gold reference. The most critical issue is completeness. Existing peer reviews often only highlight the most salient points (Logullo et al., 2025; Rethlefsen et al., 2025). Therefore, when an AI review does not overlap with the human review, it is not necessarily incorrect, but may reflect missing points in the human review (Shin et al., 2025; Matsubara, 2025). In addition, human reviewers can be inconsistent and occasionally incorrect, and may be further exacerbated by the rapidly growing submission volume (Meyerson et al., 2025; Bergstrom & Gross, 2025).

To address these issues, we propose CoCoReviewBench, which strengthens both the *completeness* and *correctness* of the reference human reviews, as shown in Figure 1. To tackle incompleteness, we construct a comprehensive two-level taxonomy of reviews with 23 subcategories. We aggregate comments from all reviewers and build per-category benchmark subsets. When the human review does not contain comments from a subcategory, we skip evaluation for that subcategory, avoiding incorrect penalization caused by incompleteness. Further, to address incorrectness, rather than relying on an LLM judge that may lack expertise, we treat discussions among the authors, other reviewers, and the meta-review, as expert annotations. When conflicts exist, at least one party is incorrect (Goldberg et al., 2023; Kargaran et al., 2025), and we use the meta-review as a high-level adjudication signal for the final decision. Finally, we construct a benchmark of 3,900 papers from ICLR and NeurIPS, providing a fine-grained and reliable analysis of AI reviewers. Our contributions are:

- We reveal incompleteness and incorrectness in human reviews, which can potentially affect the evaluation and training of AI reviewers.

- We propose CoCoReviewBench, which enables category-level evaluation and detects erroneous reference comments to mitigate the limitations of human references.

- Analysis shows that LLMs lag behind humans in correctness, are prone to hallucinations, and that reasoning models better substantiate their reviews.

## 2. Related Work

**AI Reviewers.** AI reviewers can assist and improve human review quality (Liang et al., 2024; Thakkar et al., 2025) and can produce reviews comparable to those written by humans (Weng et al., 2025). However, this raises concerns about bias (Zhu et al., 2025a), hallucination (Zhuang et al., 2025), and novelty recognition (Shin et al., 2025). To address this, one line of work builds agentic review systems that simulate the human reviewing workflow. Gao et al. (2024) decompose review writing via aspect-prompt genera-

tion, and Zhu et al. (2025b) construct multi-stage, evidence-driven analysis. D'Arcy et al. (2024) and Gao et al. (2025b) use multi-agent role specialization and aggregation, and Taechoyotin et al. (2024) extend this with multimodal retrieval. However, these pipeline-heavy systems are difficult to optimize end-to-end with existing peer review corpora, which limits performance.

Another line of work focuses on utilizing large-scale peer-review corpora. PeerRead (Kang et al., 2018) first pairs submissions with decisions and expert-written reviews. Further, Yuan et al. (2022) provide aspect annotations, MReD (Shen et al., 2022) adds meta-reviews as targets, and SEA (Yu et al., 2024) standardizes multiple reviews into unified targets. NLPeer (Dycke et al., 2023) offers a more diverse corpus with unified representations, while Tan et al. (2024) and Zhang et al. (2025) augment rebuttal traces to create a dialogue-style corpus. Building on these resources, instruction tuning (Idahl & Ahmadi, 2025; Weng et al., 2025) or reinforcement learning (Taechoyotin & Acuna, 2025; Zeng et al., 2025) is used on peer review corpora to enhance AI reviewers. However, these approaches treat human reviews as ground truth and inherit human subjectivity and occasional errors (Cortes & Lawrence, 2021; Mann et al., 2025).

Motivated by this, we construct a structured review benchmark and show that AI reviewers, like human reviewers, suffer from correctness issues and can even hallucinate, which limits their reliability.

**Evaluation of AI Reviewers.** As AI reviewers rapidly evolve, reliable evaluation becomes essential. Early evaluation treated reviewing as a prediction problem, from acceptance prediction (Kang et al., 2018) to regressing overall ratings with MSE and MAE (Zhang et al., 2025; Weng et al., 2025). However, these targets are difficult to interpret reliably due to substantial disagreement between reviewers (Cortes & Lawrence, 2021; Mann et al., 2025). For generated review text, rule-based overlap metrics are often reported against human references, e.g., BLEU (Papineni et al., 2002), ROUGE (Lin, 2004), and BERTScore (Zhang et al., 2020), but such hard matching largely rewards surface similarity and is ill-suited to peer review where valid critiques admit highly diverse expressions (Xu et al., 2025).

To move beyond overlap metrics, studies adopt the LLM-as-a-judge paradigm, enabling more flexible evaluation (Deng et al., 2024; 2025a; Idahl & Ahmadi, 2025; Gao et al., 2025a). Further work makes the assessment more diagnostic, evaluating the reviews at the opinion level rather than assigning a single overall score (Sadallah et al., 2025). One line of work does not use human references (Xu et al., 2025; Zhang & Abernethy, 2025), which is scalable but can introduce biases that inflate scores without improving substantive critique (Dorner et al., 2025; Szymanski et al., 2025). Another line refers to human reviews by matching

AI reviews with them. D'Arcy et al. (2024) use LLM to perform alignment between generated and human reviews, Chang et al. (2025) use both LLM and embedding-based similarity, and Shin et al. (2025) use labels. However, human opinions can be occasionally wrong, and the absence of a human-mentioned point does not always imply that the point is invalid for the paper (Cortes & Lawrence, 2021; Mann et al., 2025).

To address these, we introduce category-level evaluation as a compromise: we match reviews within the same category, rather than an individual opinion. We evaluate only categories covered by human reviews, resolving the incompleteness issue of using human reviews as a reference.

## 3. The Fallibility of Human Reviewers

To demonstrate the issues in human reviewing that make it unsuitable to directly serve as a gold reference, we conduct a comprehensive analysis of **3,900** papers from NeurIPS (2021–2024) and ICLR (2017–2025), which are used for the construction of CoCoReviewBench.

### 3.1. The Incompleteness of Human Reviewers

A high-quality reference for evaluation should provide a holistic evaluation of a submission. To measure completeness, based on the NeurIPS 2025 Reviewer Guidelines and the ACL Rolling Review Review Form, we build five top-level categories with 23 subcategories covering Quality, Clarity, Significance, Originality, and Policy, with construction process and definitions detailed in Appendix A.

**Sparse Single-Reviewer Coverage.** We label all human reviews with the pipeline detailed in §4, and results are shown in Figure 2. Reviewer attention is highly sparse: a single reviewer covers only 3.03 out of 5 top-level categories and 5.10 out of 23 subcategories on average, indicating that each reviewer typically comments on only a small fraction of the evaluation space. This inherent sparsity not only yields an incomplete reference during evaluation, but also presents a substantial challenge for supervised training, as models may risk inheriting this incomplete evaluation logic.

**Multi-Reviewer Gains with Residual Incompleteness.** After combining all reviewers, a paper covers 3.98 categories out of 5 and 9.23 subcategories out of 23 on average, which improves subcategory coverage by 81% compared to a single reviewer. However, the aggregated subcategory coverage is still only 40% on average, and missing coverage remains across five top-level categories, revealing that potential incompleteness still exists.

**Fine-Grained Analysis of Score Alignment.** We analyze the relationship between the reviewers' final scores and the scores for each subcategory, and find that if the average

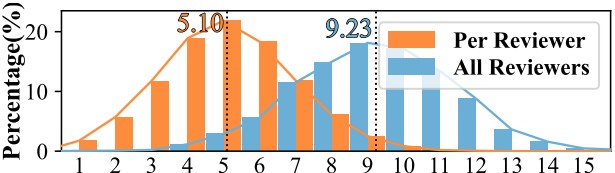

*Figure 2.* Subcategory coverage distribution of human reviewers. The x-axis shows the number of subcategories covered by one / all reviewers of each paper, and the y-axis shows their probability.

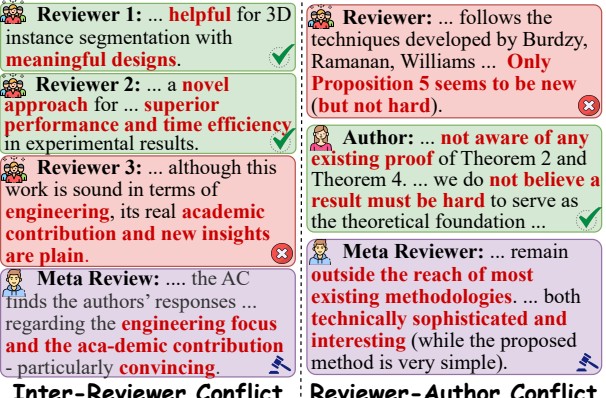

*Figure 3.* Inconsistency cases in peer reviews. We show cases of conflict both between reviewers and between reviewers and authors, with the meta-review deciding which opinion is correct.

score for the four categories: Originality-Method, Quality-Experiment, Clarity-Writing, and Quality-Comparisons, exceeds a certain threshold, we can predict an overall score of 6 or higher with over 90% recall. Further, these four categories, along with Originality-Analysis and Significance-StateOfTheArt, show the highest correlation with the final score. We further verify that the incompleteness can bias reference-based evaluation: AI opinions from categories absent in the human references receive systematically lower scores. Further conclusions are in Appendix B.1.

### 3.2. The Inconsistency of Human Reviewers

The reliability of human review is also challenged. Our analysis reveals frequent conflicts between different reviewers and between reviewers and authors, indicating that reviews may contain erroneous opinions, and reducing their reliability as gold references. Cases are shown in Figure 3.

**Inter-Reviewer Conflicts.** Over 13% of submissions exhibit a score gap of 4 or more between the highest and lowest overall scores, demonstrating the prevalence of inter-reviewer conflicts. Furthermore, to detect erroneous opinions, we aggregate reviews that discuss the same point and, when conflicts exist, we use the meta-review to designate the correct and incorrect opinions, detailed in §4. In 22.13% of papers and 7.63% of reviews, incorrectness is identified via inter-reviewer conflicts. This demonstrates that conflicts

*Table 1.* Comparison of representative peer-review datasets in terms of the supervision signals and evaluation capabilities they provide. ✓ denotes explicit support, and ✗ denotes no support.

| Resource | Atomic opinions | Fine-grained labels | Category-level evaluation | Rebuttal signal | Meta signal | Conflict detection | Incorrect annotation |
|---|---|---|---|---|---|---|---|
| PeerRead (Kang et al., 2018) | ✗ | ✓ | ✗ | ✗ | ✗ | ✗ | ✗ |
| MReD (Shen et al., 2022) | ✗ | ✓ | ✗ | ✗ | ✓ | ✗ | ✗ |
| NLPeer (Dycke et al., 2023) | ✗ | ✗ | ✗ | ✗ | ✗ | ✗ | ✗ |
| SEA (Yu et al., 2024) | ✗ | ✗ | ✗ | ✗ | ✗ | ✗ | ✗ |
| ReviewMT (Tan et al., 2024) | ✗ | ✗ | ✗ | ✓ | ✓ | ✗ | ✗ |
| Re$^2$ (Zhang et al., 2025) | ✗ | ✗ | ✗ | ✓ | ✓ | ✗ | ✗ |
| Aspects (Yuan et al., 2022) | ✓ | ✓ | ✗ | ✗ | ✗ | ✗ | ✗ |
| RottenReviews (Ebrahimi et al., 2025) | ✗ | ✓ | ✗ | ✗ | ✗ | ✗ | ✗ |
| GRE-bench (Xu et al., 2025) | ✗ | ✗ | ✗ | ✗ | ✗ | ✗ | ✗ |
| RevUtil (Sadallah et al., 2025) | ✓ | ✓ | ✓ | ✗ | ✗ | ✗ | ✗ |
| **CoCoReviewBench (Ours)** | ✓ | ✓ | ✓ | ✓ | ✓ | ✓ | ✓ |

among reviewers occur frequently, which is harmful when using them as a gold reference.

**Reviewer-Author Conflicts.** Author responses also provide valuable feedback. We determine whether authors explicitly oppose a reviewer's opinion and, in such cases, we use the meta-review to designate the correct and incorrect opinions. In 75.72% of papers and 36.76% of reviews, incorrectness is identified via reviewer–author conflicts, demonstrating the potential errors made by reviewers. Furthermore, to provide more insight, we find that most disagreements stem from misunderstandings or missed details in the paper, with cases in Appendix C.1.

**Category Distribution.** We further find that different types of errors exhibit significantly different category distributions. The three categories with the largest reviewer-to-reviewer gaps are Originality-Method, Clarity-Writing, and Quality-Experiment. In contrast, reviewer–author conflicts mainly focus on paper Quality, detailed in Appendix B.2.

## 4. CoCoReviewBench Construction

To address the limited coverage and occasional errors of human reviewers, which make them unreliable as a gold reference, we construct `CoCoReviewBench` to improve the completeness and correctness of the reference. We describe the data construction workflow in §4.2, and introduce how we evaluate AI reviewers in §4.3.

### 4.1. Design Principle

Our goal is not to synthesize reviews that are intrinsically better than humans. Instead, we annotate and filter existing human reviews, producing references that are more comprehensive and reliable than any single review. As summarized in Table 1, prior peer-review resources provide only partial supervision or evaluation signals, motivating our design to jointly improve reference completeness and correctness.

**Category-Level Benchmark.** We improve completeness in two ways. First, we aggregate reviews from multiple reviewers, increasing per-paper category coverage from 5.10 to 9.23. Besides, because even the aggregated reviews may be incomplete, we adopt a category-level metric. Rather than scoring against an overall reference that may not fully cover the paper, we construct CoCoReviewBench as a collection of category-specific sub-benchmarks and, for each, include only papers with human reviews available in that category, which prevents penalization due to missing references and provides more fine-grained insights.

**Conflict-Based Error Verification.** To improve correctness, a naive approach is to ask strong LLMs to judge errors. However, LLMs often lack strong domain expertise, making such judgments hard to trust. Peer review, however, naturally provides expert annotation signals: author rebuttals, meta-reviews, and disagreements among reviewers about the same issue. All of the above can be viewed as expert annotations about the correctness of human reviews. We treat opinion conflicts as indicators of potential errors and use these human signals to identify incorrect reviews. While this does not guarantee removing all incorrect reviews, and the meta-review is only a coarse adjudication signal rather than a perfect oracle, it is still more reliable than asking LLMs to judge correctness or using raw human reviews.

**Prior Injection via an Agent Framework.** Because the task is complex, we build an agent framework to process human reviews and manually specify what each agent does, encoding the above designs into prompts and the workflow (Wang et al., 2025). Each agent focuses on one subtask to improve quality, as shown in Figure 4. In addition, to evaluate AI reviews at the category level, we also need to classify AI reviews. To reduce cost, we distill the agent workflow into 8B models for AI review classification.

**Potential Impact.** CoCoReviewBench highlights limitations of human reviews as references for peer review evalu-

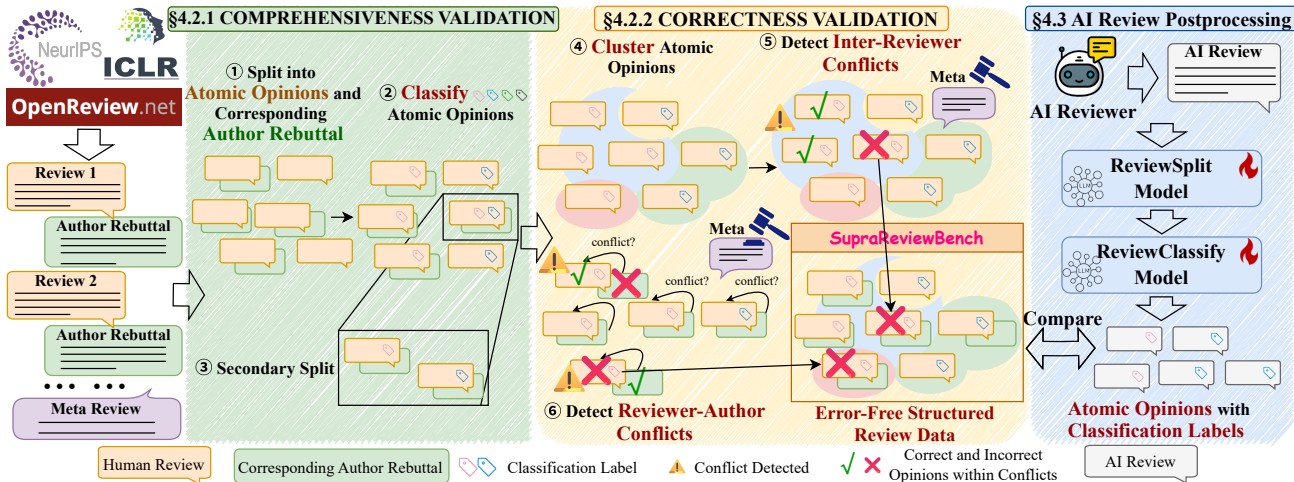

*Figure 4.* Workflow of CoCoReviewBench construction. In §4.2.1, we match each review with the corresponding author response, split it into atomic opinions, and assign labels. These results are further processed in §4.2.2: Steps 4 and 5 detect inter-review conflicts, and Step 6 detects review-author conflicts, yielding the final structured data. This structured data serves as the reference for comparison with AI reviews, which are also split and classified by the model we trained (§4.3).

ation and provides a feasible path to improve the completeness and correctness of references. Besides, for training AI reviewers, suboptimal human supervision can also be harmful. Finally, thanks to the structured annotations of our benchmark, our analysis reveals shortcomings of existing AI reviewers and motivates further improvement.

## 4.2. Benchmark Data Construction

### 4.2.1. COMPLETENESS-AWARE CLASSIFICATION

To build a category-level benchmark, we annotate labels for all human reviews by decomposing reviewer feedback into atomic opinions and classifying these opinions into 23 subcategories, as described in Appendix A.

**Review Segmentation and Classification.** Raw reviews are often long, unstructured texts. Classifying such text into multiple category groups in a single pass can easily induce hallucinations. Therefore, we split it into two steps. We first apply rule-based segmentation at newlines and semantic transitions (e.g., "However," "Furthermore"), while constraining the minimum and maximum token length of each segment. We then ask an LLM to assign sentence-level labels indicating atomic-opinion membership and the associated author response, and aggregate sentences with the same label into atomic discussion records. Then, for each atomic opinion, we use an LLM request to classify it, enabling category-level evaluation.

**Secondary Segmentation.** LLM-based segmentation may err, causing multiple opinions to be incorrectly merged into a single opinion. To address this, for atomic opinions assigned to multiple categories, we perform a secondary verification. We apply finer-grained rule-based segmentation by reducing

the allowed maximum length, reclassifying each part with an LLM, and splitting along category-change boundaries to further ensure opinion atomicity.

### 4.2.2. CORRECTNESS-AWARE CONFLICT RESOLUTION

To leverage the expert annotation signals implicit in peer review, we verify correctness from two sources, including other reviewers' assessments of the same point and the author response as expert annotation. When a conflict is detected, we use the meta-review as a high-level adjudication signal to decide which side is more reliable, and exclude the incorrect opinions from references. In addition, we remove redundant opinions to ensure that the final references contain only correct, non-duplicate opinions.

**Inter-Reviewer Redundancy and Conflict.** We aggregate discussions that target the same topic within each subcategory and retain only one as the reference to avoid redundancy and conflicts. We do so in three steps: (1) aggregate discussions about the same topic; (2) for groups containing more than one discussion, determine whether there is a conflict among opinions; (3) if a conflict exists, use the meta-review to determine the correct opinion. We implement the above with three separate LLM requests. Finally, when multiple opinions target the same topic, we only keep the longest correct opinion as the representative reference, as it typically contains the most complete justification and correct conclusion.

**Reviewer-Author Conflict.** Conflicts can also arise between reviewers and authors, i.e., when the author explicitly states that a reviewer's opinion is incorrect. We follow the same process as above, treating a review opinion and its cor-

responding author response as a group, first (1) determine whether there is an opinion conflict, and then (2) use the meta-review to decide the correct opinion. Opinions judged to be incorrect are not used as references.

### 4.2.3. CoCoReviewBench Verification

**Dataset Statistics.** We curate a large-scale dataset from NeurIPS (2021-2024) and ICLR (2017-2025), sampling 300 papers per year. Each paper has at least three independent reviewers, and at least 75% of reviews have author responses, ensuring sufficient context for verification. We use a uniform distribution across score segments: we linearly map scores to $[0, 10]$ and perform stratified sampling within each year using thresholds 3/5/7 to balance papers across four score segments. In addition, we verify OpenReview timestamps to ensure that each associated PDF corresponds to a version released earlier than the reviews. If no earlier PDF is available, we download the corresponding paper PDF from arXiv within nine months before the review release, ensuring its alignment with the reviews.

In total, we obtain **3,900** papers, with **14.1k** review comments, **134.8k** opinions, **115.9k** opinion clusters, and **108.6k** correct opinions as references, constituting the first structured dataset that includes atomic opinion segmentation, fine-grained category labels, similar-review clustering, and incorrect-opinion annotations, enabling more reliable evaluation of AI reviewers. Prompts are detailed in Appendix E.1.

**Step-Level Verification.** To demonstrate the reliability of CoCoReviewBench, at each step in the workflow, we use six different strong LLMs and, following Weng et al. (2025), apply leave-one-out cross-validation. We find relatively high agreement among the six models, demonstrating the robustness of the workflow. Meanwhile, at each step, we select the highest-scoring model to maximize the final benchmark quality. Detailed metric definitions, cross-model agreement at each step, and per-step choices are in Appendix B.3.

**Human Verification.** We randomly sample 50 papers from the final benchmark and hire PhD students in machine learning to annotate. We then compute accuracy for each annotation step, which is weighted by the number of words in its corresponding review. For review classification, 85.45% have entirely correct categories. For opinion aggregation, 93.41% are correct. For error detection caused by inter-reviewer conflicts, we achieve 81.40% accuracy. These high accuracies demonstrate the reliability of our benchmark. For error detection caused by reviewer-author conflicts, we achieve 66.83% accuracy. This accuracy is lower, because when the meta-review is overall negative, annotators tend to favor the reviewer rather than the author, especially when the meta-review does not provide a detailed justification for the specific point. For rejected papers, the annotation accuracy is only 50.03%, while for accepted pa-

pers it is 79.76%. We provide case studies in Appendix C.2 to support the reliability of our annotations.

### 4.3. AI Reviewer Evaluation

To evaluate AI reviews with our categorical benchmark, we also need to classify them. However, invoking strong LLMs to classify every AI review is expensive. Therefore, we train a smaller LLM, i.e., Qwen3-8B (Yang et al., 2025), to perform the same procedure on human reviews in §4.2.1, and use the resulting data of the previous section as training supervision. Since this stage does not need to handle author replies, it is easier, and an 8B model can perform well.

**Review Segmentation.** Here, we input AI review sentences and let the model aggregate sentences that express the same atomic opinion. For training, the dataset is small since each paper contributes only one data instance. However, the reward is dense: whether any two sentences belong to the same atomic opinion can be seen as one decision. We therefore adopt GRPO (Shao et al., 2024), generating 32 trajectories per instance to augment the data, and use the Omega Index (Lm & Cw, 1988) as the reward, which is used to evaluate clustering correctness. The final reward is:

$$R = \max(0.5, \text{OmegaIndex} + \mathbb{1}(\text{Correct Format})). \quad (1)$$

We train using results of secondary segmentation in benchmark construction, and obtain the model **ReviewSplit** for AI review segmentation.

**Review Classification.** Then, we further classify these atomic opinions. For training, the dataset is large since each opinion corresponds to one data instance. However, the reward is sparse, only correct / incorrect. If we use GRPO, results are the same within most groups, preventing effective updates. Thus, we use instruction tuning: we train the non-thinking mode of Qwen3-8B, and observe that its thinking-mode capability also improves. We finally obtain the model **ReviewClassify** for AI review classification.

**Reliability.** To validate the reliability of the above models, we first perform step-level evaluation by comparing the six strong LLMs with Qwen3-8B before and after training, and find that our trained model is substantially stronger than Qwen3-8B and approaches, or even surpasses, the performance of some strong LLMs, detailed in Appendix B.3. We further conduct human annotation and find that the model produces entirely correct classifications in 87.09% of the reviews, demonstrating the effectiveness of our model.

## 5. Experiments

Through fine-grained data cleaning, step-level quality validation, and human annotation, we construct reliable human references. Building on this, we leverage our trustworthy

references and the advantages of fine-grained categorization to conduct a more detailed analysis of the strengths and limitations of existing AI reviewers, motivating future AI reviewer development.

## 5.1. Experimental Setup

**Metrics.** We follow Tan et al. (2024) and Zhang et al. (2025), using old lexical/semantic similarity metrics, including BLEU, ROUGE-L, and BERTScore with RoBERTa-large (Liu et al., 2019), comparing the AI review with each reviewer's response and taking the maximum as the final score. However, due to the limitations of these metrics for evaluating AI reviewers, inspired by Garg et al. (2025) and Sadallah et al. (2025), we design an LLM-as-a-judge protocol tailored to category-level evaluation for a more comprehensive and reliable assessment. Specifically, we first use the 8B model we trained above to decompose AI reviews into atomic opinions and assign subcategory labels. We then compare AI reviews with human references under each category, obtaining a score from 1 to 5. Evaluation dimensions include:

- **Correctness**: Measures how well a candidate review comment aligns with the human opinions in our benchmark as gold references. By removing conflicting and highly similar comments, we retain a single most reliable atomic opinion per cluster as the reference.

- **Thoroughness**: Measures how comprehensively the comment discusses the current category, quantified by how completely the candidate covers the human reviews.

- **Grounding**: Measures how explicitly a review comment refers to a specific part of the paper and how clearly it identifies the issue with that part.

- **Verifiability**: Assesses whether a claim can be verified based on the provided reasoning, common knowledge, or external references, ensuring that the comment helps authors improve their work.

- **Clarity**: Judges how coherent, readable, and well-structured the comment is.

We evaluate only when both the human and AI provide comments in that category, assessing the correctness of existing AI comments. While the above metrics focus on *intra-category quality*, we additionally evaluate **category completeness** to measure *inter-category coverage*. Specifically, we treat the union of categories covered by all reviewers as the full score (100), and compute the ratio of the average number of categories covered per paper by the AI review to that covered by all reviewers of that paper as the final score. Only high scores on both indicate a better review. Further, **paper-level metric** uses the same prompt but scores a paper once by feeding all categories' AI reviews and human

references together, while category-level evaluation scores each category independently and averages the results.

Besides, to provide reliable reference scores, we use human performance. Concretely, we conduct **leave-one-out validation** by randomly selecting one human review as the "candidate", using the remaining reviews as references. All evaluations of AI reviews similarly use the remaining set as references. We use `GPT-5-Mini` with temperature 0 as the judge, with prompts shown in Appendix E.2. For all experiments, we randomly sample one third of the full dataset of 3,900 papers, with 100 papers per year, while preserving the original score distribution. This sampling has a negligible impact on scores, as shown in Appendix B.4.

**Evaluated Models.** We evaluate four groups of models. First, we include strong closed-source models: GPT-5.2, GPT-5-Mini, Gemini-3-Pro, and Gemini-3-Flash. Second, we evaluate general open-source reasoning models, including QwQ-32B (Team, 2025), Qwen3-8B and Qwen3-32B (Yang et al., 2025), and Nemotron-3-30B-A3B (NVIDIA, 2025). Third, we evaluate general open-source non-reasoning models, including Llama-3.3-70B, Llama-3.1-8B (Dubey et al., 2024), the non-thinking mode of Qwen3-8B, and Qwen-2.5-7B (Qwen et al., 2024). Finally, we evaluate specialized AI reviewer models, including DeepReviewer-7B and DeepReviewer-14B (Zhu et al., 2025b), CycleReviewer-8B and CycleReviewer-70B (Weng et al., 2025), OpenReviewer-8B (Idahl & Ahmadi, 2025), and SEA-E-7B (Yu et al., 2024). Among the specialized AI reviewer models, only DeepReviewer is a reasoning model; the others are non-reasoning models.

For all general-purpose models, we use the prompt from Lu et al. (2024), set the maximum generation length to 32k, and use temperature 0.6. For specialized AI reviewer models, we use the default generation settings from their original papers. If a generated review does not follow the required template, we regenerate it once. If the second generation still fails to follow the template, we keep the full output. Other settings are detailed in Appendix D.

## 5.2. Main Experiments

**Semantic Overlap over Factual Correctness of Old Metrics.** Full results are in Table 2. Under the old metrics, several of the highest-scoring systems are non-reasoning models or specialized AI reviewer models trained on human reviews. However, they even outperform the GPT-5 and Gemini-3 families, which are substantially larger and more capable. This indicates that the old metrics primarily reward semantic-level matching rather than actual correctness and usefulness. In contrast, our paper-level metrics with LLM-as-a-judge (Paper.) provide more reasonable scores, demonstrating that this is a more suitable evaluation scheme for AI reviewers.

*Table 2.* Evaluation results of various LLMs under CoCoReviewBench. Green indicates better-than-human performance, while red indicates worse-than-human performance. We mainly focus on the different behaviors of reasoning and non-reasoning models under different metrics, and their relative performance compared to humans. Metrics are defined in §5.1.

| Model | Old Metrics | | | CoCoReviewBench Categorical Metrics | | | | | | Paper. | Complete. |
|---|---|---|---|---|---|---|---|---|---|---|---|
| | BLEU | ROUGE-L | BERT. | Correct. | Thoro. | Ground. | Verify. | Clarity | Average | | |
| Human | 2.73 | 17.54 | 84.04 | 3.55 | 2.37 | 3.75 | 2.38 | 4.15 | 3.26 | 3.61 | 55.66 |
| *Strong closed-source models* | | | | | | | | | | | |
| GPT-5.2 | -1.93 | -5.06 | -1.31 | +0.36 | +0.64 | +0.92 | +0.78 | +0.32 | +0.59 | +0.90 | 84.49 |
| GPT-5-Mini | -1.90 | -5.02 | -1.29 | +0.29 | +0.58 | +0.77 | +0.53 | +0.31 | +0.48 | +0.81 | 89.97 |
| Gemini-3-Pro | -0.95 | -1.12 | -0.36 | +0.14 | +0.16 | +0.69 | +0.34 | +0.42 | +0.35 | +0.50 | 67.69 |
| Gemini-3-Flash | -1.08 | -1.10 | -0.32 | +0.09 | +0.13 | +0.69 | +0.14 | +0.41 | +0.29 | +0.44 | 68.67 |
| *Open-source reasoning models* | | | | | | | | | | | |
| Qwen3-32B | -1.30 | -2.11 | -0.55 | +0.01 | +0.12 | +0.43 | -0.15 | +0.32 | +0.14 | +0.21 | 73.17 |
| Qwen3-8B | -1.30 | -1.81 | -1.57 | -0.30 | -0.13 | -0.32 | -0.45 | -0.25 | -0.29 | -0.56 | 75.73 |
| Nemotron-3-30B-A3B | -1.51 | -2.71 | -1.09 | -0.01 | +0.12 | +0.14 | -0.29 | +0.36 | +0.07 | +0.19 | 82.93 |
| QwQ-32B | -1.38 | -2.44 | -0.63 | -0.01 | +0.13 | +0.58 | +0.02 | +0.27 | +0.19 | +0.37 | 79.83 |
| *Specialized AI Reviewer reasoning models* | | | | | | | | | | | |
| DeepReviewer-14B | -1.11 | -3.53 | -0.09 | -0.17 | +0.28 | +0.41 | +0.41 | +0.17 | +0.20 | +0.36 | 81.98 |
| DeepReviewer-7B | -1.34 | -4.43 | -0.34 | -0.22 | +0.20 | +0.18 | +0.30 | -0.08 | +0.06 | +0.07 | 89.93 |
| *Open-source non-reasoning models* | | | | | | | | | | | |
| Llama-3.3-70B | -0.41 | +1.25 | -0.21 | -0.25 | -0.23 | -0.63 | -0.78 | +0.33 | -0.30 | -0.53 | 65.49 |
| Llama-3.1-8B | -0.56 | +0.82 | -1.74 | -0.31 | -0.22 | -0.45 | -0.66 | +0.08 | -0.29 | -0.68 | 62.07 |
| Qwen3-8B-no-think | -0.87 | -0.53 | -1.10 | -0.28 | -0.10 | -0.40 | -0.58 | -0.07 | -0.29 | -0.57 | 72.28 |
| Qwen-2.5-7B | -0.75 | +0.10 | -0.64 | -0.13 | -0.06 | -0.30 | -0.59 | +0.17 | -0.18 | -0.40 | 72.40 |
| *Specialized AI Reviewer non-reasoning models* | | | | | | | | | | | |
| CycleReviewer-70B | -0.78 | +0.34 | -0.11 | -0.15 | -0.22 | -0.55 | -0.48 | +0.48 | -0.16 | -0.35 | 50.89 |
| CycleReviewer-8B | -0.61 | +1.21 | -0.27 | -0.16 | -0.14 | -0.70 | -0.48 | +0.30 | -0.21 | -0.54 | 47.92 |
| OpenReviewer-8B | -0.46 | +0.14 | -0.12 | -0.08 | -0.17 | -0.52 | -0.31 | +0.37 | -0.12 | -0.30 | 48.63 |
| SEA-E-7B | -1.63 | -1.80 | -1.37 | -0.53 | -0.44 | -1.62 | -1.11 | -0.38 | -0.79 | -1.33 | 51.54 |
| Average | -1.10 | -1.54 | -0.73 | -0.09 | +0.04 | -0.04 | -0.19 | +0.20 | -0.02 | -0.08 | 70.31 |

**Clearer but Less Correct and Thorough AI Reviews.** Further, our category-level multi-dimensional evaluation enables a finer-grained analysis. For consistency with human references (Correct.) and how thorough AI reviews discuss this category (Thoro.), these two dimensions remain relatively weaker overall: the stronger closed-source models can exceed human reviewers, but many other models still score lower than human reviewers. In contrast, their clarity scores are consistently higher than those of human reviews. This suggests that current AI reviewer systems still underperform on core evaluation criteria.

**Human-Level Grounding and Verifiability in Reasoning Models.** Grounding and verifiability measure whether a model can clearly provide the source and rationale for its reviews. The results show that reasoning models have a significant advantage and are comparable to, or even better than, human reviewers. This indicates that modern LRMs can generate well-supported reviews, often in more detail than humans. In contrast, non-reasoning models perform substantially worse, highlighting the necessity of reasoning.

**Broader but Shallower Category Coverage in AI Reviews.** Across categories, we find that AI reviews often cover more categories than a single human review (Complete.), especially reasoning models, but fewer than the aggregation of all reviewers, with their scores consistently below 100. Further, within each category, AI reviews provide no better thorough analysis within each category on average, especially on non-thinking models. This shows that, compared to humans, AI reviews have broader category coverage but insufficient depth, and their breadth still has room for improvement. Further, we build a simple agent framework that can ensure full category coverage without reducing correctness, with potential to synthesize more comprehensive references, as detailed in Appendix B.5.

**Meta-Review Adjudication Analysis** Our conflict-based filtering relies on meta-reviews as high-level adjudication signals. We therefore further analyze whether meta-reviews provide reliable guidance in conflict scenarios. We consider papers with explicit conflict signals and evaluate the original meta-review on four 1–5 dimensions using the same LLM-as-a-judge setting as in §5.1. *Adjudication Clarity* measures whether the meta-review states a clear stance on the main disputed issue. *Groundedness* measures whether this stance is supported by reviewer comments and discussion context. *Verifiability* measures whether the judgment is concrete enough to be checked. *Conflict Coverage* measures whether the meta-review addresses the major disputed points rather than only part of them.

*Table 3.* Quality of meta-reviews in conflict scenarios. Strong-conflict papers are those with at least five incorrect opinions.

| Conflict | Clarity | Ground. | Verify. | Cover. | Avg. |
|---|---|---|---|---|---|
| 1–4 | 3.32 | 2.95 | 2.93 | 2.58 | 2.94 |
| ≥5 | 3.60 | 3.27 | 3.23 | 2.85 | 3.24 |

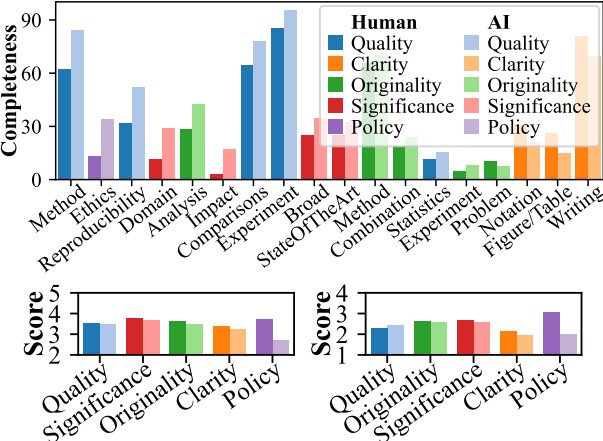

*Figure 5.* Category-wise breakdown of the results in Table 2. **Top:** the percentage of a model's opinions that cover each category. **Bottom:** the category-wise correctness (left) and thoroughness scores (right) (1–5). All categories are ordered from left to right by the score of AI minus that of humans, so categories on the left indicate stronger AI performance.

We define strong-conflict papers as papers with at least five detected incorrect opinions. As shown in Table 3, strong-conflict papers receive higher scores than weak-conflict papers across all four dimensions, with the average score increasing from 2.94 to 3.24. Moreover, 73.1% of strong-conflict papers have a four-dimensional average score of at least 3, suggesting that meta-reviews usually provide a meaningful high-level signal about the main dispute when conflicts are substantial. However, the scores remain far from perfect, especially on Conflict Coverage. This confirms that meta-reviews should be treated as coarse adjudication signals rather than fine-grained ground truth for every detailed dispute.

### 5.3. Categorical Analysis of AI Reviewers

To diagnose where the gaps in Table 2 come from, we break down the results by category to provide further insights.

**Broader Major-Category Coverage.** The upper panel in Figure 5 reports category completeness: we compute the percentage of a model's opinions that cover each category at least once for each paper. AI reviews outperform humans on Quality-related categories, as well as on some categories that humans rarely cover, indicating that AI reviewers pay more attention to Quality-related categories. Surprisingly, although we do not provide images as input, the coverage for

Figure-related categories is also high, suggesting potential hallucination risks, which we analyze later.

**Strong Quality Performance but Weak Policy and Clarity.** The bottom of Figure 5 shows the scores for correctness and thoroughness. High scores on both indicate strong performance. At the major-category level, Quality is the strongest dimension for AI reviewers and is already close to, or slightly above, the human baseline. Originality and Significance are more mixed but remain relatively close to human performance on average. In contrast, the clearest weaknesses appear in Clarity and especially Policy, where AI reviewers lag behind humans in both correctness and thoroughness. This suggests that the main remaining gaps are no longer uniformly spread across all categories, but are concentrated in a few major dimensions. Further analysis of scores in other dimensions is in Appendix B.6.

**Hallucination Risks for AI Reviewers.** In all the experiments, we use text-only inputs. However, we find that language models occasionally generate figure-related opinions. Although the probability is low—fewer than 0.05 opinions per paper—this occurs in every model, indicating hallucinations. Further, we treat the opinions judged as incorrect as references and find that the gap between humans and AI in correctness narrows substantially, demonstrating that AI may also fit such erroneous opinions, leading to potential hallucinations, with cases and analysis in Appendix B.7.

## 6. Conclusion

We introduce `CoCoReviewBench`, a completeness- and correctness-oriented benchmark for evaluating AI reviewers. It enables category-level evaluation that avoids unfair penalties from missing human references, and improves correctness by leveraging reviewer–author–meta discussions to filter erroneous opinions. Experiments show that traditional metrics can be misleading, and that correctness remain relatively weaker dimensions for AI reviewers, especially outside the strongest closed-source models. We also advocate using reasoning models as AI reviewers, which perform better on most metrics. We hope our reliable benchmark and further insights facilitate AI reviewer development.

**Limitations.** First, conflict-based filtering only detects errors that surface as explicit disagreements, and can also carry sentiment framing, causing erroneous opinions to be missed when responses are absent or overly mild. However, this may still be more reliable than LLM-based judgments, which often lack timely domain expertise. Besides, our evaluation requires additional computation to segment and categorize AI reviews. We mitigate this by distilling the pipeline into 8B models. Finally, beyond providing more complete feedback, LLMs should also be able to identify the most critical reviews, which remains largely unexplored.

## Acknowledgments

This work is supported by the Zhongguancun Academy (Grant No.s C20250203). Derek F. Wong was supported in part by the Science and Technology Development Fund of Macau SAR (Grant Nos. FDCT/0007/2024/AKP, EF2024-00185-FST), the UM and UMDF (Grant Nos. MYRG-GRG2024-00165-FST-UMDF, MYRG-GRG2025-00236-FST), the Tencent AI Lab Rhino-Bird Research Program (Grant No. EF2023-00151-FST), the Dr. Stanley Ho Medical Development Foundation (Grant No. SHMDF-AI/2026/001), and the National Natural Science Foundation of China (Grant No. 62266013).

## Impact Statement

Our work facilitates the development and reliable evaluation of AI reviewer systems in multiple dimensions.

- We highlight limitations of human reviews, including incomplete category coverage and the presence of conflicts, which indicate erroneous opinions and make human reviews suboptimal as both training data and evaluation references. This issue has often been overlooked in prior AI reviewer development. We call for greater attention to this limitation, which may ultimately propel AI reviewers to surpass human performance.

- We propose `CoCoReviewBench`, which addresses these issues with a more complete and fine-grained evaluation protocol, improving the reliability of AI reviewer evaluation. We hope it can provide a more reliable and standardized benchmark for future AI reviewer development and enable fair comparisons across different AI reviewer systems.

- Our fine-grained analysis shows that current AI reviewers still lag behind humans in correctness and thoroughness, especially for non-Quality categories, and can exhibit hallucinations. Meanwhile, reasoning models achieve stronger grounding and verifiability, motivating the use of reasoning models for future AI reviewer development.

Finally, for ethical aspects, LLMs may exhibit racial and gender biases, so we strongly recommend users assess potential biases before applying the models in specific contexts. Additionally, due to the difficulty of controlling LLM outputs, users should be cautious of issues arising from hallucinations.

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

# A. Peer Review Taxonomy

To comprehensively evaluate the content of peer reviews and construct categorical sub-benchmarks, we develop a hierarchical taxonomy consisting of 5 top-level categories with 23 subcategories. This finer granularity enables more precise diagnosis of AI reviews, and facilitates more detailed review analysis that yields additional insights.

## A.1. Hierarchical Taxonomy Definitions

To build a reliable and comprehensive set of categories, we largely follow the NeurIPS 2025 Reviewer Guidelines[1] and the ACL Rolling Review Review Form[2]. We define five top-level categories as follows: the first four are from NeurIPS 2025, while the last category is mentioned in the ACL Rolling Review Review Form but not covered by the first four categories.

**Quality**: Is the submission technically sound? Are claims well supported (e.g., by theoretical analysis or experimental results)? Are the methods used appropriate? Is this a complete piece of work or work in progress? Are the authors careful and honest about evaluating both the strengths and weaknesses of their work?

**Clarity**: Is the submission clearly written? Is it well organized? (If not, please make constructive suggestions for improving its clarity.) Does it adequately inform the reader? (Note that a superbly written paper provides enough information for an expert reader to reproduce its results.)

**Significance**: Are the results impactful for the community? Are others (researchers or practitioners) likely to use the ideas or build on them? Does the submission address a difficult task in a better way than previous work? Does it advance our understanding/knowledge on the topic in a demonstrable way? Does it provide unique data, unique conclusions about existing data, or a unique theoretical or experimental approach?

**Originality**: Does the work provide new insights, deepen understanding, or highlight important properties of existing methods? Is it clear how this work differs from previous contributions, with relevant citations provided? Does the work introduce novel tasks or methods that advance the field? Does this work offer a novel combination of existing techniques, and is the reasoning behind this combination well articulated? As the questions above indicate, originality does not necessarily require introducing an entirely new method. Rather, work that provides novel insights by evaluating existing methods, or demonstrates improved efficiency, fairness, etc., is also equally valuable.

**Policy**: Encompasses policy- or compliance-related concerns such as ethics, data/privacy compliance, anonymity rules, plagiarism, licensing, and broader impact. These issues often require checking adherence to conference or legal policies and may involve ethical considerations.

To enable fine-grained evaluation, training, and analysis, we further decompose these 5 dimensions into 23 fine-grained subcategories, forming our hierarchical taxonomy. This granular decomposition supports a more rigorous and systematic assessment of review coverage, ensuring that each critical aspect of a scientific manuscript is explicitly evaluated. The detailed definitions are presented in Table 4.

## A.2. Detailed Construction Process

To construct the subcategories, we primarily rely on data mining via GPT-5 in agent mode to extract subcategories that are as comprehensive as possible from large-scale existing data. Concretely, using the reviewer guidelines above as exemplars, we first prompt the model to retrieve reviewer guidelines from venues such as NeurIPS, ICLR, and ACL Rolling Review to form an initial taxonomy draft. We generate multiple candidate versions with this procedure. We then evaluate each version on a small development set by asking GPT-5 and Gemini-2.5-Pro to classify review comments under the candidate taxonomy, compute classification consistency (inter-model agreement), and select the version with the highest consistency.

Next, we further refine the above taxonomy by having GPT-5 agent mode collect large volumes of OpenReview reviews and attempt to categorize them under the current taxonomy. This process helps identify missing and frequently confused categories, prompting further taxonomy refinement and the addition of boundary rules to disambiguate borderline cases. We also feed cases where GPT-5 and Gemini-2.5-Pro disagree on the test set back into the refinement loop as additional signals. Finally, we finalize 23 distinct subcategories. After the category set is fixed, we continue refining and adding boundary rules to reduce model confusion and produce the final prompt for classification.

---

[1] https://neurips.cc/Conferences/2025/ReviewerGuidelines
[2] https://aclrollingreview.org/reviewform

*Table 4.* Complete taxonomy of CoCoReviewBench categories and subcategories.

| Category | Subcategories and Descriptions |
|---|---|
| **Quality** | **Quality-Method** (Methodological Soundness): Evaluates whether the proposed algorithms, models, or system architectures are technically correct and free of conceptual or implementation errors. 
 **Quality-Experiment** (Experimental Design & Evaluation): Assesses the adequacy of the experimental setup: choice of datasets, baselines, evaluation metrics, ablation studies, and statistical tests. 
 **Quality-Reproducibility** (Reproducibility & Implementation Details): Considers whether the submission provides enough detail (e.g., hyperparameters, code availability, dataset splits) to replicate the work and whether the implementation follows best practices. 
 **Quality-Comparisons** (Comparisons to Prior Work): Evaluates whether the paper sufficiently compares against relevant prior work and state-of-the-art methods. 
 **Quality-Statistics** (Statistical Rigor & Validation): Evaluates whether statistical analyses (e.g., significance tests, confidence intervals) are properly applied and whether reported improvements are statistically meaningful. |
| **Clarity** | **Clarity-Writing** (Writing, Terminology & Algorithm Presentation): Covers overall writing quality and organization, precision of terminology and key concepts, and the clarity of algorithm presentations (code snippets, pseudocode, workflow diagrams). 
 **Clarity-Notation** (Notation & Mathematical Explanation Clarity): Considers whether mathematical notation is consistent, well defined and explained. 
 **Clarity-Figure/Table** (Figures & Visual Aids Clarity): Evaluates whether figures, tables, plots, and visualizations are legible, properly labeled, and aid understanding. |
| **Significance** | **Significance-Broad** (Broad Research Impact): Evaluates whether the work addresses a broadly important problem or advances understanding in a way that is likely to influence multiple research areas. 
 **Significance-Domain** (Domain/Applied Impact): Assesses the significance of the contribution for a specific application domain (e.g., NLP, robotics, healthcare). 
 **Significance-StateOfTheArt** (Improvement Over State of the Art): Focuses on the magnitude and importance of improvements relative to current state-of-the-art methods. 
 **Significance-Impact** (Real-World & Societal Impact): Considers the potential practical and societal ramifications of the work, including benefits, risks, fairness, environmental impacts and accessibility. |
| **Originality** | **Originality-Problem** (Novel Problem Formulation): Comments on the introduction of a new problem, task, or dataset. Includes innovative problem definitions that highlight previously unaddressed challenges. 
 **Originality-Method** (Novel Methodology or Algorithm): Assesses whether the paper introduces a genuinely new algorithmic approach or architectural design rather than a minor tweak of existing methods. 
 **Originality-Analysis** (Novel Analysis or Insights): Pertains to original theoretical insights or analyses that deepen understanding of existing methods or phenomena. 
 **Originality-Experiment** (Novel Experimental Setup or Data): Evaluates whether the paper proposes new experimental setups, benchmarks, evaluation protocols or collects new datasets. 
 **Originality-Combination** (Creative Combination of Existing Methods): Considers whether the paper combines well-known techniques in an original way and whether the rationale behind the combination is compelling. 
 **Originality-NegativeResults** (Negative Results or Critical Assessments): Covers papers that provide critical evaluations, ablations or negative results showing limitations of existing methods. |
| **Policy** | **Policy-Ethics** (Ethics & Responsible AI Compliance): Addresses ethical concerns, including fairness, bias, harms to marginalized populations, dual use and whether the authors appropriately discuss and mitigate ethical risks. 
 **Policy-DataPrivacy** (Data Usage & Privacy Compliance): Considers whether the data used in the paper comply with privacy regulations and licensing terms (e.g., consent, personally identifiable information). 
 **Policy-Anonymity** (Double-Blind or Anonymity Violations): Concerns whether the submission inadvertently reveals author identities or institutional affiliations, violating double-blind review policies. 
 **Policy-Plagiarism** (Plagiarism & Dual Submission): Addresses issues of plagiarism, self-plagiarism, or dual submission to multiple venues. 
 **Policy-BroaderImpact** (Broader Impact & Societal Considerations): Evaluates whether the authors complete required checklists and discuss the broader impact of their work on society. |

# B. Supplementary Analysis in CoCoReviewBench

## B.1. Categorical Analysis of Human Peer Review in CoCoReviewBench

To provide additional insights, we analyze which fine-grained review dimensions in our 23-subcategory taxonomy most strongly align with the final overall score. Let $i \in \{1, \ldots, N\}$ index reviews and $j \in \{1, \ldots, M\}$ index subcategories (with $M=23$). Let $y_i$ denote the final overall score for review $i$. For each pair $(i, j)$, we collect the reviewer text $t_{ij}$ corresponding to subcategory $j$ (i.e., all review sentences annotated with $j$, concatenated as input). We then use `GPT-5-Mini` to map $t_{ij}$ to an ordinal subcategory score $x_{ij} \in \{1, 2, 3, 4\}$ with fixed criteria (1: Poor, 2: Fair, 3: Good, 4: Excellent). If the reviewer does not mention subcategory $j$, then $x_{ij}$ is missing. We write $m_{ij} = \mathbb{I}[x_{ij} \text{ observed}]$ and define $\mathcal{I}_j = \{i : m_{ij} = 1\}$.

### B.1.1. EXPERIMENT SETUP

We use complementary metrics that capture association strength, multivariate attribution under correlated dimensions, and decision relevance under a high-score criterion. Throughout this section, all statistics are computed only from each reviewer's initial review in CoCoReviewBench to avoid inconsistencies across multi-round discussions.

**Coverage.** Because human reviews are sparse, we report two mention-based coverage rates for each category/subcategory $j$. **Single Cov.** is the percentage of individual reviews $i$ that cover $j$, i.e., $m_{ij} = 1$. **All Cov.** is the percentage of papers (3,900 total) for which at least one reviewer covers $j$ in their initial review, i.e., $\sum_i m_{ij} > 0$. By construction, All Cov. is an upper bound on Single Cov. and quantifies how much coverage improves when aggregating multiple reviewers.

**Correlation.** To measure how strongly each subcategory score varies with the final score when it is actually discussed, we compute **Pearson** correlation $r_j$ (Pearson, 1895) and **Spearman** rank correlation $\rho_j$ (Spearman, 1904) using only the observed subset $\mathcal{I}_j$. Define the subset means

$$\bar{x}_j = \frac{1}{|\mathcal{I}_j|} \sum_{i \in \mathcal{I}_j} x_{ij}, \qquad \bar{y} = \frac{1}{|\mathcal{I}_j|} \sum_{i \in \mathcal{I}_j} y_i, \tag{2}$$

and compute

$$r_j = \frac{\sum_{i \in \mathcal{I}_j}(x_{ij} - \bar{x}_j)(y_i - \bar{y})}{\sqrt{\sum_{i \in \mathcal{I}_j}(x_{ij} - \bar{x}_j)^2}\sqrt{\sum_{i \in \mathcal{I}_j}(y_i - \bar{y})^2}}, \qquad \rho_j = r\left(\mathrm{rank}_{\mathcal{I}_j}(x_{ij}), \, \mathrm{rank}_{\mathcal{I}_j}(y_i)\right), \tag{3}$$

where $r(\cdot, \cdot)$ denotes the Pearson correlation and $\mathrm{rank}_{\mathcal{I}_j}(\cdot)$ ranks values over $i \in \mathcal{I}_j$.

**Ridge Regression.** We further fit a linear model to estimate a stable multivariate signal, since univariate correlations can be inflated or suppressed when subcategories co-occur. To use all reviews and produce a fixed-length feature vector, we mean-impute missing subcategory scores by the observed subcategory mean:

$$\tilde{x}_{ij} = \begin{cases} x_{ij}, & m_{ij} = 1, \\ \bar{x}_j, & m_{ij} = 0, \end{cases} \qquad \tilde{\mathbf{x}}_i = (\tilde{x}_{i1}, \ldots, \tilde{x}_{iM})^\top \in \mathbb{R}^M. \tag{4}$$

We then use Ridge regression with $\ell_2$ regularization (Hoerl & Kennard, 1970), which can mitigate coefficient instability under collinearity:

$$\min_{\beta_0, \beta} \frac{1}{N} \sum_{i=1}^N \left(y_i - \beta_0 - \tilde{\mathbf{x}}_i^\top \beta\right)^2 + \alpha\|\beta\|_2^2. \tag{5}$$

We tune $\alpha$ via $k$-fold cross-validation ($k=5$), searching over $10^{-4}$ to $10^4$.

**Mentioned vs. Missing Effect.** Association metrics conflate score magnitude and the event of being discussed. To explicitly quantify whether merely mentioning a subcategory is informative, we compare the mean final score between reviews where $x_{ij}$ is present versus missing:

$$\Delta_j = \mathbb{E}[y_i \mid m_{ij} = 1] - \mathbb{E}[y_i \mid m_{ij} = 0]. \tag{6}$$

**Composite Score Search (F1).** To obtain a simple decision-oriented signal that combines multiple dimensions, we exhaustively search over all subcategory combinations of size 1 to 4. For a candidate set $S \subset \{1, \ldots, M\}$, define the

*Table 5.* Coverage and association of each review subcategory score with the final overall score, sorted by Pearson. Single Cov. is the percentage of individual reviews that mention the subcategory; All Cov. is the percentage of papers for which at least one reviewer mentions it. Pearson $r$ and Spearman $\rho$ are computed on the observed subset. Ridge $\beta$ is the coefficient from a multivariate Ridge model with mean-imputation. Missing $\Delta$ is the mean overall-score difference between mentioned vs. missing. Cell colors indicate within-column relative magnitude (green: larger; red: smaller).

| Category | Single Cov. | All Cov. | Pearson | Spearman | Ridge | Missing $\Delta$ |
|---|---|---|---|---|---|---|
| Quality | 97.00 | 99.92 | 0.4752 | 0.4849 | 0.9303 | -0.432 |
| Originality | 70.67 | 95.31 | 0.4524 | 0.4487 | 0.6367 | -0.022 |
| Clarity | 80.16 | 98.49 | 0.3432 | 0.3324 | 0.4258 | +0.122 |
| Significance | 47.55 | 83.44 | 0.3213 | 0.3168 | 0.3879 | +0.160 |
| Policy | 7.72 | 20.85 | 0.1857 | 0.1711 | 0.2420 | -0.327 |
| Originality-Method | 45.22 | 80.28 | 0.4410 | 0.4471 | 0.4945 | -0.294 |
| Quality-Experiment | 81.60 | 97.28 | 0.4291 | 0.4357 | 0.5127 | -0.199 |
| Originality-Analysis | 26.00 | 50.90 | 0.4167 | 0.4025 | 0.5113 | +0.314 |
| Significance-StateOfTheArt | 18.79 | 45.28 | 0.4066 | 0.4042 | 0.3685 | +0.054 |
| Clarity-Writing | 73.34 | 96.97 | 0.3705 | 0.3599 | 0.3792 | +0.096 |
| Quality-Comparisons | 55.65 | 91.21 | 0.3608 | 0.3738 | 0.4325 | -0.517 |
| Quality-Method | 53.87 | 87.15 | 0.3433 | 0.3458 | 0.3252 | -0.086 |
| Originality-Combination | 9.49 | 25.46 | 0.3213 | 0.3159 | 0.3592 | +0.000 |
| Originality-Problem | 5.56 | 15.54 | 0.3203 | 0.2997 | 0.2596 | -0.026 |
| Policy-Anonymity | 0.26 | 0.90 | 0.3153 | 0.3373 | 0.0372 | -1.339 |
| Originality-Experiment | 3.95 | 8.59 | 0.3129 | 0.3006 | 0.2428 | +0.307 |
| Significance-Broad | 25.39 | 58.85 | 0.2823 | 0.2923 | 0.3102 | +0.234 |
| Quality-Reproducibility | 32.31 | 67.15 | 0.2802 | 0.2745 | 0.2910 | +0.031 |
| Quality-Statistics | 6.73 | 19.95 | 0.2635 | 0.2695 | 0.1712 | -0.302 |
| Clarity-Notation | 26.89 | 59.97 | 0.2552 | 0.2592 | 0.3208 | -0.088 |
| Originality-NegativeResults | 1.04 | 3.41 | 0.2522 | 0.2276 | 0.1167 | +0.444 |
| Clarity-Figure/Table | 23.00 | 55.77 | 0.2398 | 0.2368 | 0.1741 | +0.096 |
| Policy-DataPrivacy | 0.55 | 1.62 | 0.2365 | 0.2431 | 0.0434 | -0.203 |
| Significance-Impact | 2.57 | 7.54 | 0.1891 | 0.1749 | 0.1583 | +0.320 |
| Significance-Domain | 10.50 | 29.00 | 0.1694 | 0.1624 | 0.1804 | +0.126 |
| Policy-Plagiarism | 0.31 | 1.03 | 0.1685 | 0.1051 | 0.0686 | -1.469 |
| Policy-Ethics | 5.60 | 15.41 | 0.1450 | 0.1443 | 0.2007 | -0.279 |
| Policy-BroaderImpact | 1.26 | 4.15 | 0.1171 | 0.0912 | -0.0025 | -0.060 |

composite score for review $i$ as

$$s_i(S) = \frac{1}{|S|} \sum_{j \in S} \tilde{x}_{ij}. \tag{7}$$

We then choose a threshold $\tau$ that maximizes F1 for predicting $\mathbb{I}[y_i \geq 6]$ ("HIGH"),

$$\tau^{\mathrm{HIGH}}(S) \in \arg\max_\tau \ \mathrm{F1}\big(\mathbb{I}[s_i(S) \geq \tau], \ \mathbb{I}[y_i \geq 6]\big), \tag{8}$$

and analogously for predicting $\mathbb{I}[y_i \leq 4]$ ("LOW"),

$$\tau^{\mathrm{LOW}}(S) \in \arg\max_\tau \ \mathrm{F1}\big(\mathbb{I}[s_i(S) \leq \tau], \ \mathbb{I}[y_i \leq 4]\big). \tag{9}$$

### B.1.2. WHICH REVIEW DIMENSIONS MOST ALIGN WITH FINAL DECISIONS

**Incomplete Coverage for a Single Reviewer.** The full results are shown in Table 5. The coverage columns (Single Cov. vs. All Cov.) show that a single reviewer does not consistently cover all core aspects: Significance is mentioned in fewer than half of individual reviews (47.55%), which is unexpectedly low for a primary evaluation criterion. Policy is also rarely mentioned, which is more acceptable because policy issues are typically triggered only in a small minority of papers. Aggregating all reviewers per paper substantially increases coverage (e.g., Significance rises to 83.44%), but still leaves 16.56% of papers without any explicit Significance discussion, indicating that aggregation helps yet remains insufficient for completeness.

**Stable Important Category across Association and Multivariate Attribution.**   Pearson and Spearman correlations produce nearly identical top rankings. The largest associations are Originality-Method, Quality-Experiment, Originality-Analysis, and Significance-StateOfTheArt, followed by Clarity-Writing and Quality-Comparisons. Ridge regression yields the largest coefficients on the same subcategories. The Ridge-predicted overall score for review $i$ is

$$\hat{y}_i = 0.5127\,\tilde{x}_{i,\text{Quality-Experiment}} + 0.5113\,\tilde{x}_{i,\text{Originality-Analysis}} + 0.4945\,\tilde{x}_{i,\text{Originality-Method}} + 0.4325\,\tilde{x}_{i,\text{Quality-Comparisons}}$$
$$+ 0.3792\,\tilde{x}_{i,\text{Clarity-Writing}} + 0.3685\,\tilde{x}_{i,\text{Significance-StateOfTheArt}} + 0.3592\,\tilde{x}_{i,\text{Originality-Combination}} + ...... \tag{10}$$

where $\tilde{x}_{i,\cdot}$ are the mean-imputed subcategory scores as defined above. Overall, the categories with the largest influence on the overall score remain consistent across these complementary metrics, underscoring their importance.

**Missingness Effects Show that Absence is not Neutrality.**   Mentioned-vs-missing comparisons reveal that certain categories are raised mainly in negative contexts. The largest negative shifts occur for Policy-Plagiarism and Policy-Anonymity. Although these are rare events, they indicate that policy-related concerns can substantially depress the expected overall score. In contrast, Originality-NegativeResults and Significance-Impact show positive shifts when present, suggesting that explicitly articulated novelty and impact statements accompany higher scores. This cautions against treating "not mentioned" as an implicit negative label in sparse reviews.

**Composite Signals Predict High Scores Well, but Low Scores Remain Hard.**   Using the composite-score search over up to four dimensions, the best "HIGH" classifier for $y \geq 6$ achieves F1= 0.7827 (P= 0.6870, R= 0.9092) at $\tau$=1.6803 with metrics Clarity-Writing, Originality-Method, Quality-Comparisons, and Quality-Experiment. Top-performing sets are highly consistent and repeatedly include the same core dimensions identified by correlation and Ridge. In contrast, predicting "LOW" ($y \leq 4$) is substantially more difficult: the best F1 is only 0.3888 despite searching the same space, indicating that low-score decisions are not well captured by a small mean-aggregated subset of subcategory sentiments, while high-score decisions are comparatively more predictable.

### B.1.3. EVALUATION BIAS FROM INCOMPLETE HUMAN REFERENCE CATEGORIES

We further test whether incomplete human category coverage introduces bias when human reviews are directly used as references for evaluating AI reviewers. For each paper, we split the generated AI review into two disjoint subsets according to the categories covered by the human references: *category-consistent* opinions, whose categories are covered by the human reference, and *category-inconsistent* opinions, whose categories are absent from the human reference. We then evaluate the two subsets separately against the same paper-level human reference, using the same paper-level LLM-as-a-judge protocol as in Section 5.1. We keep only papers where both subsets are non-empty and have comparable length, requiring their character-length ratio to be within 2x. A negative delta means that the category-inconsistent subset receives a lower score than the category-consistent subset.

As shown in Table 6, category-inconsistent opinions are consistently scored lower across all four models. The largest drops appear in Correctness and Thoroughness, which depend most directly on agreement with the reference, while the drops in Grounding, Verifiability, and Clarity are smaller. This suggests that the performance gap is not solely caused by lower general review quality, but also reflects a systematic evaluation bias: once human reviews are treated as the reference universe, valid comments from categories missing in the human reference can be disadvantaged. Further, even after aggregating all reviewers, human reviews cover only 9.23 out of 23 subcategories on average. It shows that treating human-mentioned content as a complete reference can create a sparse and biased evaluation target.

*Table 6.* Score differences between category-inconsistent and category-consistent AI opinions under paper-level evaluation. Negative values indicate that opinions from categories absent in human references are scored lower.

| Model | $\triangle$Correct. | $\triangle$Thoro. | $\triangle$Ground. | $\triangle$Verify. | $\triangle$Clarity | $\triangle$Avg. |
|---|---|---|---|---|---|---|
| Llama-3.1-8B-Instruct | -0.16 | -0.27 | -0.21 | -0.07 | -0.07 | -0.15 |
| QwQ-32B | -0.37 | -0.44 | -0.23 | -0.27 | -0.19 | -0.30 |
| DeepReviewer-7B | -0.16 | -0.27 | -0.09 | -0.02 | -0.09 | -0.13 |
| Qwen3-32B | -0.37 | -0.43 | -0.37 | -0.28 | -0.24 | -0.34 |
| Average | -0.27 | -0.35 | -0.23 | -0.16 | -0.15 | -0.23 |

## B.2. Category Distribution of Reviewer Incorrectness

We further analyze where reviewer incorrectness occurs in our fine-grained taxonomy. We assign each identified incorrect opinion in our benchmark to one of the subcategories, and report the percentage of papers that contain at least one incorrect opinion of that category. The results are shown in Figure 6.

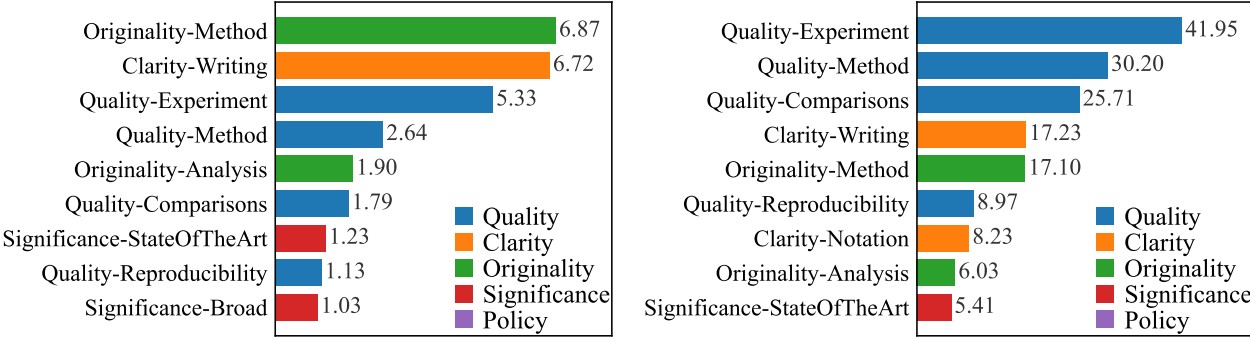

*Figure 6.* Subcategory distribution of reviewer incorrectness. **Left:** inter-reviewer conflicts. **Right:** reviewer–author conflicts. Colors indicate the corresponding major categories.

**Inter-Reviewer Conflicts.** The largest inter-reviewer gaps concentrate on Originality-Method, Clarity-Writing, and Quality-Experiment, followed by Quality-Method and Originality-Analysis. This indicates that substantial inter-reviewer disagreements are relatively widespread and can arise across many different categories.

**Reviewer-Author Conflicts.** Reviewer-author conflicts are dominated by Quality-related subcategories, especially Quality-Experiment, Quality-Method, and Quality-Comparisons. We also observe notable conflicts on Clarity-Writing and Originality-Method. Overall, author rebuttals most often challenge reviewers on concrete technical details such as experimental setups, methodological assumptions, and the adequacy of comparisons, suggesting that a large portion of reviewer errors stem from the correctness of the paper.

### B.3. Step-Level Verification and Model Selection for CoCoReviewBench

Our CoCoReviewBench construction process is divided into multiple steps. To verify the quality of each step, and select the most suitable LLM for each stage of the CoCoReviewBench construction workflow, we conduct a preliminary study that measures cross-model consistency and uses it to select the best-performing models for our pipeline. Through mutual verification, we address the lack of a gold reference, enabling more reliable validation.

Specifically, our workflow involves eight LLM calls. First, in §4.2.1, we segment each review into atomic opinions (Review Segmentation), then classify the atomic opinions (Review Classification), and further split atomic opinions that contain multiple categories (Secondary Segmentation), totaling three LLM calls. Then, in §4.2.2, we cluster comments that discuss the same topic (Review Clustering) and then separately examine conflicts between reviewers (Inter-Reviewer) and between reviewers and authors (Reviewer-Author). Each examination consists of two steps: first identifying whether a conflict exists within a comment group (Conflicts-Identification), and then, based on the meta-review, determining which opinions in the conflicting group are correct or incorrect (Conflicts-Resolution), totaling five LLM calls.

#### B.3.1. EXPERIMENTAL SETUP

**Candidate Models.** We evaluate six leading LLMs: GPT-5, GPT-5-Mini, Gemini-2.5-Pro, Gemini-2.5-Flash, DeepSeek-R1 (Guo et al., 2025), and DeepSeek-V3 (Liu et al., 2024a), which span a wide range of cost and performance profiles.

**Leave-One-Out Cross-Validation.** We sample 180 post-2020 papers that are not included in our benchmark, with 20 papers per year serving as a validation set. For each pipeline step, we run all six models on the same inputs, which are generated by the outputs of the best-performing model selected in the previous step. Because no gold answers exist for these tasks, we follow Weng et al. (2025) and use a leave-one-out cross-validation strategy: for each input, we treat the consensus derived from the other five models as a pseudo-ground truth and compute a consistency score by comparing the held-out model's output against this consensus. We then rank candidate models for each step to select the best model.

**Metrics.** For tasks that integrate multiple components, including Review Segmentation and Review Clustering, we assess cross-model agreement by computing the pairwise **Omega Index** between models. For multi-class classification tasks, including Review Classification and Secondary Segmentation, we measure **Consistency** by the average of (i) the proportion of model pairs that produce exactly identical labels and (ii) the proportion that produce partially identical label sets. Because hard voting is often unreliable in the above setting, we use a soft-label scheme: for each instance, we treat the mean prediction of the other five models as a soft label and compute each model's agreement with this soft label, and we calculate this by the average agreement across model pairs. For binary classification tasks, including Inter-Reviewer Conflict Identification and Resolution and Reviewer-Author Conflict Identification and Resolution, we instead compute the **F1 score** using a majority-vote pseudo-gold label: an option selected by at least four of the six models serves as the gold label, and we assign a score of 0.5 in the event of a tie (3 out of 6). These gold labels are preserved across all models.

#### B.3.2. SELECTION RESULTS

Table 7 presents the cross-validation results for Review Segmentation, Review Classification, Secondary Segmentation and Review Clustering. For Review Segmentation, GPT-5's superior performance secures its selection. In Review Classification, GPT-5-Mini is chosen after balancing comparable capabilities from GPT-5 and Gemini-2.5-Pro against cost considerations. GPT-5-Mini also excels in Secondary Segmentation, making it the assigned model for this step as well. Finally, DeepSeek-R1's optimal performance in Review Clustering determines its adoption for that task.

As shown in Table 8, for Inter-Reviewer Conflict Identification and Resolution, DeepSeek-R1 delivers the best performance, which is used for this step. For Reviewer-Author Conflict Identification, to balance performance and cost, we calculate the consistency of the intersection between Gemini-2.5-Flash and GPT-5-Mini judgments, achieving a consistency value of 87.67, which outperforms GPT-5's 84.56 at a lower cost. For Reviewer-Author Conflict Resolution, DeepSeek-R1 shows the best performance and is used for this step. Overall, the results above show high consistency across different models, demonstrating the robustness and effectiveness of our design.

Based on these results, we select a hybrid pipeline using the best-performing model for each specific sub-task, keeping the total construction cost to approximately $1,000 for 3,900 papers while maximizing reliability.

After selecting the generative models for Review Segmentation and Review Classification, we use these models to generate corresponding segmentations and classifications on the training set, and train Qwen3-8B with them, obtaining ReviewSplit

*Table 7.* The cross-validation results among six models for Review Segmentation, Review Classification, Secondary Segmentation and Review Clustering.

| Model | Review Segmentation | Review Classification | Secondary Segmentation | Review Clustering |
|---|---|---|---|---|
| GPT-5 | **75.07** | 75.35 | 80.16 | 70.73 |
| GPT-5-Mini | 69.44 | 75.41 | **81.36** | 70.83 |
| Gemini-2.5-Pro | 72.62 | **75.59** | 80.15 | 65.00 |
| Gemini-2.5-Flash | 71.30 | 74.81 | 80.82 | 71.32 |
| DeepSeek-R1 | 72.90 | 74.73 | 78.65 | **71.58** |
| DeepSeek-V3 | 73.29 | 69.15 | 70.65 | 70.92 |

*Table 8.* The cross-validation results among six models for Inter-Reviewer Conflict Identification and Resolution as well as Reviewer-Author Conflict Identification and Resolution.

| Model | Inter-Reviewer | | Reviewer-Author | |
|---|---|---|---|---|
| | Conflicts-Identification | Conflicts-Resolution | Conflicts-Identification | Conflicts-Resolution |
| GPT-5 | 85.38 | 88.12 | **84.56** | 67.68 |
| GPT-5-Mini | 91.80 | 85.26 | 78.76 | 83.04 |
| Gemini-2.5-Pro | 88.09 | 85.71 | 76.04 | 87.87 |
| Gemini-2.5-Flash | 90.23 | 92.78 | 78.40 | 86.54 |
| DeepSeek-R1 | **93.33** | **94.00** | 71.86 | **89.15** |
| DeepSeek-V3 | 79.45 | 82.69 | 67.78 | 85.24 |

*Table 9.* The cross-validation results among eight models for Review Segmentation and Review Classification, newly adding Qwen3-8B and our trained ReviewSplit / ReviewClassify.

| Model | Review Segmentation | Review Classification | Average |
|---|---|---|---|
| GPT-5 | 76.41 | 72.91 | **74.66** |
| GPT-5-Mini | 74.28 | **73.81** | 74.05 |
| Gemini-2.5-Pro | 75.12 | 72.95 | 74.04 |
| Gemini-2.5-Flash | 75.61 | 72.33 | 73.97 |
| DeepSeek-R1 | **76.91** | 71.91 | 74.41 |
| DeepSeek-V3 | 74.75 | 67.54 | 71.15 |
| Qwen3-8B | 69.43 | 67.59 | 68.51 |
| ReviewSplit / ReviewClassify | 73.24 | 71.62 | 72.43 |

and ReviewClassify, respectively. Here, compared with our previous evaluation using human-written reviews, we instead use LLM-generated reviews as inputs to test its ability to process AI reviews. Specifically, we randomly sample 25 reviews generated from our benchmark with DeepReviewer-7B, CycleReviewer-8B, OpenReviewer-8B, and SEA-E-7B, forming a 100-review test set. We present the cross-verification results for Qwen3-8B and our trained models in Table 9. The results show that our trained models achieve significant performance improvements over Qwen3-8B and even surpass the 671B DeepSeek-V3 model, demonstrating that our trained models generalize well to AI reviews and achieve sufficiently strong performance.

### B.3.3. ERROR PROPAGATION ANALYSIS

The above analysis validates each step through cross-model consistency. We further analyze whether residual upstream errors propagate to downstream adjudication errors. Our human verification is not a set of isolated module validations, but an end-to-end validation of the final benchmark's correctness. To better understand error propagation, we decompose the pipeline into two correctness paths: (1) review classification → reviewer-author conflict adjudication (Cls→RA), and (2) review classification → opinion grouping → reviewer-reviewer conflict adjudication (Cls→Grp→RR). For each path, we compare downstream adjudication errors under clean and imperfect upstream conditions.

First, the upstream structured steps are relatively reliable. On 50 human-annotated papers, review classification achieves 85.45% accuracy, and opinion grouping achieves 93.41% accuracy. Second, residual upstream errors do propagate, but they are not the dominant source of downstream errors. As shown in Table 10, in the reviewer-author path, the weighted downstream adjudication error rate increases from 31.41% under clean upstream conditions to 47.45% under imperfect

upstream conditions. This gives a propagation penalty of 16.04 percentage points. In the reviewer-reviewer path, the imperfect-upstream subset contains only two entries, so we do not report a separate weighted imperfect-upstream error rate. Nevertheless, in both paths, most downstream error words still come from entries with clean upstream outputs: upstream-imperfect entries account for only 15.73% and 3.17% of downstream error words in the two paths, respectively.

*Table 10.* Error propagation analysis under clean and imperfect upstream conditions. Clean denotes the share of review words under clean upstream conditions. CleanErr and ImpErr denote downstream weighted error rates under clean and imperfect upstream conditions. ErrFromUp denotes the proportion of downstream error words arising from entries with upstream issues.

| Chain | Clean | CleanErr | ImpErr | $\Delta$ | ErrFromUp |
|---|---|---|---|---|---|
| Cls$\rightarrow$RA | 89.01% | 31.41% | 47.45% | +16.04% | 15.73% |
| Cls$\rightarrow$Grp$\rightarrow$RR | 73.15% | 24.63% | – | – | 3.17% |

Third, downstream errors are not uniformly distributed across papers. After aggregating adjudication entries at the paper level, the average correctness ratio is 66.67%, but the distribution is highly uneven: 48.00% of papers are entirely correct, 18.00% are entirely incorrect, and only 12.00% fall into the middle 40–60% correctness range. This suggests that downstream errors are concentrated in a subset of difficult papers rather than evenly affecting all papers. Therefore, the propagation penalty should not be interpreted as arising solely from step-level error transmission. Part of the residual error also reflects intrinsic task difficulty, since papers with more ambiguous comments and discussion contexts are harder to adjudicate.

Overall, residual errors do propagate along both correctness paths, but the effect is limited. Given the relatively high accuracy of the upstream structured steps and the small fraction of downstream errors attributable to upstream issues, error propagation is not the only source, nor the dominant source, of final benchmark errors.

### B.4. Score Consistency Between the 1300-Sample and the Full 3900 Set

Our evaluation uses a 1,300-example subset sampled from the full 3,900-example CoCoReviewBench. To verify that this subsampling does not materially change the aggregate scoring behavior, we run experiments on the full set of 3,900 papers with four representative models and the human leave-one-out score, and compare the model-level dimension averages computed on the 1,300-sample subset and on the full 3,900 set in Table 11. We additionally report the human leave-one-out score as a calibration row.

Let $M$ denote the set of four models (Qwen3-8B, QwQ-32B, Llama-3.1-8B, and DeepReviewer-7B) and $D$ the set of five metrics. For each model $m \in M$ and metric $d \in D$, we summarize the discrepancy using mean absolute error (MAE):

$$\text{MAE} = \frac{1}{|M||D|} \sum_{m \in M} \sum_{d \in D} |s_{m,d}^{(3900)} - s_{m,d}^{(1300)}|. \tag{11}$$

We obtain an overall $\text{MAE} = 0.026$ across 20 model–metric pairs. In particular, the changes in the overall averages are minimal before and after subsampling, and the relative ranking among the four AI models remains unchanged. This indicates that the sampled 1,300-set scores closely match the full 3,900-set scores, demonstrating the robustness of using only the 1,300-set for system-level evaluation.

*Table 11.* Overall scores under CoCoReviewBench categorical metrics on the full 3,900-paper set and the 1,300-paper subset. For each model, MAE is the mean absolute error between the 1,300-subset and 3,900-set scores, averaged over the five metric dimensions.

| Model | CoCoReviewBench Categorical Metrics (1300) | | | | | | CoCoReviewBench Categorical Metrics (3900) | | | | | | MAE |
|---|---|---|---|---|---|---|---|---|---|---|---|---|---|
| | Correct. | Thoro. | Verif. | Ground. | Clarity | Average | Correct. | Thoro. | Verif. | Ground. | Clarity | Average | |
| Human | 3.55 | 2.37 | 2.38 | 3.75 | 4.15 | 3.26 | 3.50 | 2.31 | 2.31 | 3.72 | 4.17 | 3.22 | 0.043 |
| Qwen3-8B | 3.25 | 2.24 | 1.94 | 3.43 | 3.90 | 2.97 | 3.22 | 2.22 | 1.86 | 3.39 | 3.89 | 2.93 | 0.036 |
| QwQ-32B | 3.54 | 2.50 | 2.41 | 4.33 | 4.43 | 3.45 | 3.55 | 2.51 | 2.40 | 4.35 | 4.45 | 3.46 | 0.015 |
| Llama-3.1-8B | 3.24 | 2.14 | 1.72 | 3.30 | 4.23 | 2.98 | 3.22 | 2.14 | 1.66 | 3.27 | 4.26 | 2.96 | 0.028 |
| DeepReviewer-7B | 3.33 | 2.56 | 2.68 | 3.93 | 4.07 | 3.32 | 3.31 | 2.56 | 2.64 | 3.93 | 4.12 | 3.31 | 0.024 |

*Table 12.* Overall scores of baseline AI Reviewers and our trained models. Green indicates better-than-human performance, while red indicates worse-than-human performance.

| Model | Correctness | Thoroughness | Verifiability | Grounding | Clarity | Average |
|---|---|---|---|---|---|---|
| Human | 3.55 | 2.37 | 2.38 | 3.75 | 4.15 | 3.26 |
| GPT-5.2 | +0.36 | +0.64 | +0.78 | +0.92 | +0.32 | +0.59 |
| Gemini-3-Pro | +0.14 | +0.16 | +0.34 | +0.69 | +0.42 | +0.35 |
| Gemini-3-Flash | +0.09 | +0.13 | +0.14 | +0.69 | +0.41 | +0.29 |
| Qwen3-32B | +0.01 | +0.12 | -0.15 | +0.43 | +0.32 | +0.14 |
| DeepReviewer-14B | -0.17 | +0.28 | +0.41 | +0.41 | +0.17 | +0.20 |
| ReviewMerge | -0.08 | +0.39 | +0.12 | +0.43 | +0.31 | +0.21 |
| ReviewSingle | -0.07 | +0.39 | +0.13 | +0.45 | +0.33 | +0.23 |
| ReviewAgent | +0.30 | +1.09 | +0.96 | +1.01 | -0.35 | +0.58 |

## B.5. Better Reference for AI Reviewer Evaluation

Human peer reviews are often incomplete or incorrect, which can lead to suboptimal AI reviewers. In this appendix, we therefore explore a separate synthesis setting that aims to generate stronger references for benchmarking reviewer systems; this is distinct from the main benchmark-construction pipeline, which only annotates and filters existing human reviews. Specifically, in the main text, we only evaluate categories where both human and AI comments are available, which can lead to incomplete coverage. We therefore explore data synthesis to construct more comprehensive and reliable references beyond the limitations of existing human reviews.

**Completeness-Aware Agent Workflow.** To tackle the incomplete category coverage of human reviews, we train categorical agents to address this issue, similar to how we use categorical sub-benchmarks in the main text. Each agent generates comments for a single subcategory, and we concatenate all agents' outputs into the final review, which naturally ensures comprehensive category coverage. If a subcategory lacks human comments for a paper, we skip training that agent to avoid learning missing-category biases. We use papers from ICLR 2017–2025 and NeurIPS 2021–2024 that are not included in CoCoReviewBench, and filter out papers containing over 23k tokens, yielding 34k papers and 130k review comments. We then split and classify them into subcategory-level comments using ReviewSplit and ReviewClassify. We treat overly short category comments as missing and ignore Figure-related categories, resulting in 321k category–paper pairs for training.

**Correctness-Aware Training Data Construction.** Directly concatenating different reviewers' comments of the same category can introduce stylistic/formatting inconsistencies, redundancy, and errors, while running the full CoCoReviewBench filtering pipeline is too costly. We therefore train ReviewClean to deduplicate and remove incorrect opinions using intermediate outputs from the CoCoReviewBench construction process. This task is relatively difficult, so our goal is to enable the model to merge opinions from multiple reviews into fluent text, while deleting repeated and incorrect opinions as much as possible, without pursuing overly high accuracy.

To build ReviewClean training data, we use Qwen3-8B to rewrite the cleaned, non-duplicated, and non-conflicting reviews in each category into fluent text. We then pair the noisy version before deduplication/conflict removal with this rewritten cleaned version as training data, so that ReviewClean learns to remove duplicated/conflicting opinions and fix formatting. We follow the ReviewClassify training setup, but train for two epochs with a batch size of 128. Finally, we apply ReviewClean to produce category-level comments for instruction tuning of our agent workflow.

**Experimental Setup.** We train this model using an instruction-tuning setup similar to ReviewClassify, yielding the ReviewAgent model. We also consider a simple baseline that concatenates all human reviewer comments as references and trains a model to generate as comprehensive comments as possible, referred to as ReviewMerge. In Table 12, we additionally report ReviewSingle and several strong single-model AI reviewers for reference. Across all experiments, we use the same settings: batch size 256, a maximum of 1,200 steps, and a maximum sequence length of 32k, with other settings consistent with ReviewClassify. At inference time, we use the non-reasoning mode, with hyperparameters kept consistent with the main experiments.

**Improved References Generated by Our Model.** The results are shown in Table 12. ReviewAgent substantially outperforms the simpler ReviewMerge and ReviewSingle baselines across most metrics, and also exceeds both human references and strong single-model baselines across most metrics, although GPT-5.2 still remains slightly stronger on average. Our approach makes 22 model calls to generate category-specific comments and therefore produces more content. Nevertheless, from the perspective of constructing more reliable references, these results suggest that structured multi-stage synthesis can improve substantially over direct merge-based baselines, while still leaving room for future work on stylistic quality and efficiency. For reliability and provenance, in the main text, we still use human reviews as references, which are often considered more trustworthy.

## B.6. Categorical Score Analysis

Our benchmark includes multiple review categories and multiple evaluation dimensions, enabling multi-perspective evaluation and fine-grained analysis. In the main text, we have already mentioned that the AI Reviewer performs well on Quality-related categories, maintaining high correctness and analytical depth. Here, we further report scores along additional dimensions and conduct a finer-grained analysis. The full results are shown in Figure 7.

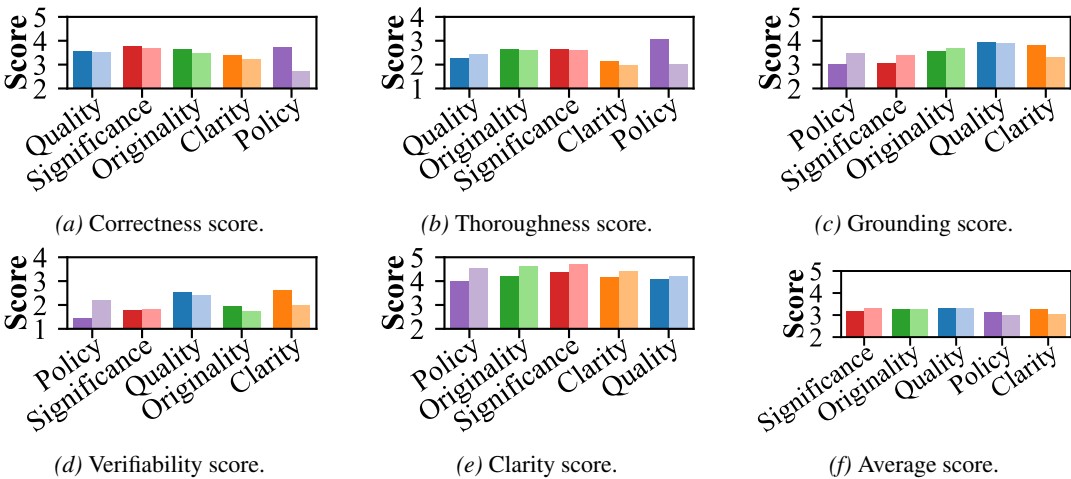

*(a)* Correctness score.    *(b)* Thoroughness score.    *(c)* Grounding score.

*(d)* Verifiability score.    *(e)* Clarity score.    *(f)* Average score.

*Figure 7.* Categorical metric scores across major review categories in CoCoReviewBench. Panels (a–f) correspond to correctness, thoroughness, grounding, verifiability, clarity, and the average score. The x-axis lists the major review categories. Within each panel, categories are ordered by the AI–Human score gap. Colors indicate the corresponding major review categories, with blue corresponding to Quality, orange to Clarity, green to Originality, red to Significance, and purple to Policy.

**Clarity remains the most consistent weakness for AI reviewers.**    Across major review categories, the Clarity category shows the most consistent negative gap for AI reviewers. It ranks near the weaker end in correctness, grounding, verifiability, and especially in the final average score. This indicates that, although AI reviews themselves are usually fluent, current AI reviewers are still less effective at judging whether the paper is clearly written, well organized, and sufficiently explained.

**AI policy reviews are weak in correctness and thoroughness, while human policy reviews are weak in formal dimensions.**    The Policy category exhibits a distinct pattern. In correctness and thoroughness, AI reviewers perform worse than humans, showing that policy-related judgments remain unreliable and insufficiently deep. However, the pattern is different for more formal dimensions such as grounding, verifiability, and clarity. In these dimensions, human Policy comments are themselves relatively weak, and AI reviewers can obtain comparable or even better relative scores. One possible reason is that human reviewers often raise policy concerns briefly and procedurally, whereas AI reviewers tend to provide more explicit explanations. Therefore, the relatively better formal scores of AI reviewers in Policy should not be interpreted as stronger policy judgment, since the core dimensions of correctness and thoroughness remain weak.

**AI reviewers can match humans in grounding and verifiability for Quality.**    As discussed in the main text, Quality is already one of the relatively stronger categories for AI reviewers in terms of correctness and thoroughness. Here, we further find that this advantage also extends to the evidence-related dimensions. For the Quality category, human reviewers show strong grounding and verifiability. This is expected because Quality-related comments usually concern concrete technical issues. Such comments often have clearer evidence in the paper and are easier to verify. Importantly, AI reviewers can largely match humans on grounding and verifiability in this category, suggesting that current AI reviewers can provide feedback with comparable specificity and checkability.

**Originality and Significance maintain relatively strong performance.**    Originality and Significance maintain relatively favorable performance across the evaluation dimensions. They do not show the severe correctness and thoroughness weakness observed in Policy, nor the broad degradation observed in Clarity. In the average-score panel, Originality and Significance achieve the top two AI–Human gaps among the five major categories, indicating that current AI reviewers are comparatively stronger at high-level contribution and impact assessment than at policy compliance or detailed clarity diagnosis.

## B.7. Hallucination in AI Reviewers

**Case Study of AI Reviewer Hallucination in Figure.**   Although our evaluation uses text-only inputs, i.e., models do not have access to figures, we occasionally observe figure-specific comments in AI reviews. We refer to this phenomenon as figure-related hallucination: the model produces confident, concrete critiques that implicitly assume visual evidence that is not available in the input. While such opinions are rare in aggregate, they are concerning because they mimic plausible peer review feedback and can mislead authors and readers when presented without provenance.

Figure 8 shows three representative cases. Left: Gemini-3-Pro criticizes the caption for mentioning a "z-axis" and questions consistency with "2D" plots, despite not seeing the figure. Middle: Qwen3-32B asserts that the qualitative analysis in "Figure 2/3" lacks diverse examples, framing a visual deficiency without access to the underlying visuals. Right: DeepReview-14B claims that "Figure 8" is poorly documented and lacks clear axis labels and scaling information. Collectively, these cases suggest that insufficiently filtered training data can lead models to generate mismatched yet persuasive critiques.

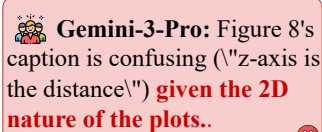

👥 **Gemini-3-Pro:** Figure 8's caption is confusing (\"z-axis is the distance\") **given the 2D nature of the plots.**. ⊗

👥 **Qwen3-32B**: Qualitative analysis **(e.g., Figure 2/3) lacks diverse examples**, reducing confidence in the meta-knowledge graph's interpretability. ⊗

👥 **Deepreview 14B:** The figures in the paper are also poorly documented, particularly **Figure 8, which lacks clear axis labels and scaling information.** This makes it difficult to interpret the results and to verify the authors' claims. ⊗

*Figure 8.* Cases of hallucinated AI reviews. In each case, providing appropriate review feedback requires understanding the figure. However, none of the AI reviewers in these examples are given the figure as input.

**Higher Similarity to Incorrect Opinions.**   Further, we examine whether broader hallucination phenomena exist across a wider range of categories. CoCoReviewBench labels opinions as correct or incorrect and constructs a set of incorrect opinions. Higher similarity to these incorrect opinions can be regarded as hallucination, i.e., incorrectly fitting the erroneous opinions in the data. Specifically, compared with the main text, where we only use correct opinions as the gold reference (w/ Correct), here we use only incorrect opinions as the reference (w/ Incorrect) and report the final Correctness score, which represents the similarity between the AI review and the corresponding reference. To reduce interference, we ignore the Clarity-Figure/Table subcategory since we do not input images during evaluation.

The results are shown in Table 13. Humans' agreement with incorrect opinions decreases, demonstrating the validity of incorrect-opinion extraction. On the AI side, the pattern is not uniform across all models, but several strong systems, including GPT-5.2, GPT-5-Mini, Gemini-3-Pro/Flash, and Qwen3-32B, show higher similarity to incorrect opinions than humans, and the average row is also slightly above the human baseline. This indicates a measurable hallucination risk if incorrect opinions are used as references for training or evaluation (Deng et al., 2023; Liu et al., 2024b; Ke et al., 2025), even though some weaker models remain less aligned with these incorrect references.

*Table 13.* Correctness results using incorrect or correct human opinions as references.

| Model | Correctness | |
|---|---|---|
| | **w/ Incorrect** | **w/ Correct** |
| Human | 3.08 | 3.55 |
| GPT-5.2 | +0.72 | +0.34 |
| GPT-5-Mini | +0.48 | +0.28 |
| Gemini-3-Pro | +0.14 | +0.15 |
| Gemini-3-Flash | +0.10 | +0.09 |
| Qwen3-32B | +0.09 | +0.01 |
| Qwen3-8B | -0.06 | -0.30 |
| Nemotron-3-30B-A3B | +0.15 | -0.02 |
| QwQ-32B | +0.07 | 0.00 |
| DeepReviewer-14B | -0.16 | -0.17 |
| DeepReviewer-7B | -0.17 | -0.22 |
| Llama-3.3-70B | -0.23 | -0.23 |
| Llama-3.1-8B | -0.33 | -0.28 |
| Qwen3-8B-no-think | -0.11 | -0.26 |
| Qwen-2.5-7B | -0.06 | -0.11 |
| CycleReviewer-70B | -0.02 | -0.14 |
| CycleReviewer-8B | -0.01 | -0.16 |
| OpenReviewer-8B | -0.05 | -0.07 |
| SEA-E-7B | -0.25 | -0.50 |
| Average | +0.02 | -0.09 |

## C. Case Study

### C.1. Causes of Reviewer-Author Conflicts

To further analyze the causes of reviewer–author conflicts, we randomly select 50 human-labeled conflict instances and find that they can be categorized into three types: misunderstandings, overlooked details, and inaccurate prior assessments. We present three cases to illustrate how these different types of discrepancies manifest in the peer review process.

**Case 1: Misunderstanding.** Misunderstanding is defined as a scenario where the reviewer misunderstands the main contribution, logic, or goal of the paper. As shown in Figure 9 (Left), the reviewer misinterprets the paper's contribution as an incremental reuse of existing components. In the rebuttal, the authors clarify the core positioning of the research and explain why such a reductionist characterization fails to capture the key technical innovation. Crucially, the meta-review acknowledges the contribution of the study and supports the authors' clarification. This case demonstrates a conflict driven by interpretive misalignment rather than a substantive flaw in the research itself.

**Case 2: Missed Details.** Missed details is defined as instances where the reviewer misses specific technical details, formulas, parameter settings, or experimental results that are explicitly present in the paper. Figure 9 (Middle) illustrates this scenario, where the questions raised and information requested by the reviewer actually exist in the original submission. The authors subsequently point out the specific locations of the relevant content. This case highlights a conflict resulting from the reviewer's oversight of details, leading to redundant critiques.

**Case 3: Inaccurate Prior Assessment.** Inaccurate prior assessment is defined as a situation where the reviewer criticizes the work based on subjective bias, unsubstantiated premises, or standard practices, leading the authors to respond under incorrect assumptions. Unlike the previous two types, which indicate that the reviewer does not read the paper carefully, this type reflects a reviewer's hallucinated prior assessment, resulting in an erroneous review. As shown in Figure 9 (Right), the reviewer confidently claims that the theorems follow "standard techniques" from specific prior works, questioning the paper's novelty. In the rebuttal, the authors refute this claim, noting that these specific results do not appear in the existing literature. The meta-review acknowledges the novelty and practical impact of the method. This case illustrates a conflict that arises when a reviewer's baseless external assumption leads to an inaccurate assessment of novelty.

*Figure 9.* Cases of reviewer-author conflicts. **Left:** Misunderstanding case. **Middle:** Missed details case. **Right:** Inaccurate prior assessment case.

## C.2. Analysis of CoCoReviewBench Reliability

For error detection caused by Reviewer-Author conflicts, we achieve 66.83% accuracy in human verification. This accuracy is notably lower than other annotation steps, and exhibits a substantial disparity between rejected and accepted papers: 50.03% versus 79.76%. To investigate the underlying causes of this discrepancy, we conduct case-specific analyses on rejected papers where disagreement is more pronounced.

We analyze items in rejected papers where human annotators classify the Reviewer-Author conflicts in our dataset as incorrect. As shown in Figure 10, we present two representative cases in which the authors provide factually grounded rebuttals that effectively refute the reviewers' technical misconceptions. Crucially, the corresponding meta-review fails to adjudicate these specific disputes, instead citing general reasons to support the rejection. Influenced by the negative attitude of the meta-review toward the paper, human annotators label our dataset as erroneous, implying that the reviewers' critiques contained no errors. This reveals a tendency for human annotators to align with critical assessments when the meta-review is overall negative, leading them to overlook objectively valid rebuttals, particularly when the meta-review lacks detailed justification for the specific contested point.

These observations suggest that the lower accuracy rate is primarily attributable to bias, whereby the negative verdict retroactively influences annotators' judgment of the rebuttal's validity. This indicates that the annotation accuracy for this step is actually higher, demonstrating both the effectiveness of our method and the inherent difficulty of the task.

> **Reviewer: I wonder about the novelty compared with other works** in Graph convolution. From my viewpoint, **the Graph Conv/ Messaging pass network is also locally permutation-equivariant**.

> **Author: It is not true** that Graph/Conv/ Message passing is locally permutation-equivariant. Message passing is permutation equivariant to global permutations of the graph, **but each local update on sub-graphs is permutation invariant** ... It has been acknowledged by many works that **global permutation equivariant models scale badly** and therefore we aimed to maintain the expressivity while improving the scalability...

> **Meta Reviewer:** This paper presents a GNN architecture that adopts locally permutation-equivariant constructs, which has better scalability ... **All reviewers unanimously recommended rejection**, and the main issues are the clarity and writing...

> **Reviewer:** The rebuttal comments state that "we did not tune them and simply used the same hyperparameter settings in the original papers where these models were described." This seems to reinforce the notion that this **is not a fundamentally different model, but one intimately related to existing papers**.

> **Author:** We believe **there is a misunderstanding** regarding your response on hyperparameter tuning. We referred to "hyperparameters for training" taken the same as the original paper. These **do not include hyperparameters for sparse reparameterization** (which is the subject of this paper and we did tune those). We **do not believe this is related to the novelty of our method** ... If anything, using the exact hyperparameters for training as in the original papers only **strengthens our method**...

> **Meta Reviewer:** the reviewers still had concerns about that the work appears to be incremental relative to SET, and that the differences in performance between the two models were not very large... **the AC ultimately decide to recommend rejection**...

*Figure 10.* Cases of bias in human annotation under negative meta-review.

## D. Additional Experiment Setup

**Human References.** For the old metrics including BLEU, ROUGE-L, and BERTScore, we follow Zhang et al. (2025) to compute the similarity between an AI review and each human reviewer's comments. Here, consistent with prior work, we only consider the reviewer's first review as references. However, further replies are also valuable. Therefore, in our benchmark, we additionally include further replies as references, which may contain more correct or deeper opinions after discussion with the authors. Moreover, among all opinions that discuss the same topic, we keep only the longest correct opinion as the representative. If all opinions in a group are incorrect, we drop the group. This ensures that the references are non-redundant and correct. For our paper-level LLM-as-a-judge evaluation, we use all such opinions as references, while categorical evaluation considers only the opinion groups assigned to the corresponding subcategory. Finally, to ensure a fair comparison with AI reviewers, for each evaluated human reviewer, we keep only the first review and apply the ReviewClassify model for classification in the same way as AI Reviewers. In addition, for the full-score reference in the category completeness (Complete.) metric, i.e., the category coverage aggregated from all reviewers, we also use the ReviewClassify model for re-classification. In other cases, we still use the CoCoReviewBench annotations for the references. Besides, in all the experiments above, we use text-only inputs. Accordingly, when constructing the human references, we remove reviews that require figures.

**Training Details of ReviewSplit and ReviewClassify.** ReviewSplit is trained using GRPO without KL loss, directly training the model's reasoning mode with a temperature of 0.7, and with both the maximum input length and maximum output length set to 12k tokens (Deng et al., 2025b; Rao et al., 2025). The reward is defined in the main text. The training data include the split results of CoCoReviewBench after the Secondary Split step. Here, we use full-parameter training for 2 epochs, generating 32 trajectories per question and 128 trajectories per batch, with a learning rate of 1e-6. ReviewClassify is trained using instruction tuning (Wei et al., 2022), training its non-reasoning mode, while testing uses the reasoning mode. It is fine-tuned with low-rank adaptation (LoRA; Hu et al., 2022), using a learning rate of 2e-5, a LoRA rank of 16, and a LoRA alpha of 32, applying LoRA to all linear layers. The training data include the opinion classification results of CoCoReviewBench before and after the Secondary Split step, enabling the model to learn how to handle inputs with different numbers of atomic opinions. We train for one epoch with a batch size of 256.

# E. Prompts

## E.1. Prompts for CoCoreviewBench Construction

We present all the prompts used in our paper in §4 in the following tables, including:

- **Prompts for Review Segmentation:** The prompt is used to split review comments and author responses into atomic opinions when constructing the benchmark. We first segment the review comments and author responses into sentence-level units according to rules, then use this prompt to aggregate the sentence-level opinions based on content, resulting in multiple atomic opinions and corresponding author responses.

- **Prompts for Review Classification:** The prompt is used to assign category labels to the atomic opinions obtained from the previous step when constructing the benchmark.

- **Prompts for Secondary Segmentation:** The prompt is used to review the labeling results above. We find that Review Segmentation may aggregate multiple opinions into one block, resulting in non-atomic opinions with excessive labels. To address this, for segmented opinions with multiple labels, we attempt further splitting. Specifically, we reassign the most suitable label to each sentence within the preliminarily segmented opinions from the available multiple labels. If different parts of an opinion belong to different label groups, we split it into separate opinions and determine the label of each secondary-split opinion based on its sentence labels, thereby resolving this issue.

- **Prompts for Review Clustering:** The prompt is used to further cluster atomic opinions that share the same category label based on whether they discuss the same specific content (e.g., several atomic opinions all discussing the correctness of the same formula).

- **Prompts for Inter-Reviewer Conflict Identification:** The prompt is used to identify whether conflicts exist within atomic opinion groups derived from Review Clustering, i.e., to identify whether reviewers' views on the specific content under discussion are consistent.

- **Prompts for Inter-Reviewer Conflict Resolution:** The prompt is used to determine which atomic opinions are correct and which are incorrect within the conflicting atomic opinion groups obtained from Inter-Reviewer Conflict Identification, and the meta-review is introduced to assist in this judgment.

- **Prompts for Reviewer-Author Conflict Identification:** The prompt is used to identify whether the Author explicitly disagrees with a specific atomic review from the Reviewer.

- **Prompts for Reviewer-Author Conflict Resolution:** The prompt is used to determine whose opinion is correct when the Reviewer-Author Conflict Identification step identifies atomic comments explicitly opposed by the author, and the meta-review is introduced to assist in this judgment.

- **Prompts for ReviewSplit Model:** The purpose of this prompt is similar to "Prompts for Review Segmentation", but it only segments review comments, excluding author responses.

- **Prompts for ReviewClassify Model:** The purpose of this prompt is similar to "Prompts for Review Segmentation", but it only segments review comments, excluding author responses.

## Prompts for Review Segmentation

```
You are given a peer-review discussion split into {num} parts, each prefixed
with "ID: {{id}}".  Each part is either a reviewer comment or an author
reply.

# GOAL
Assign a numeric label to EVERY part so that all parts about the same
discussion Point share the same label.
- Point:  one atomic strength, weakness, or question raised by a reviewer.
- Labels start at 1 ("Label 1", "Label 2", ...).  Use the same number for
the reviewer's Point and all replies addressing it.
- Use "Label N/A" only for purely polite text with no technical or content
value (e.g., "Thank you") or for pure paper summaries of the paper.
- If a part is a citation (e.g., "[1] ...") but clearly supports or
questions a known Point, use that Point's label.  Do not use "Label N/A".

# PROCEDURE
1.  Identify atomic Points in reviewer comments and create a new label for
each.
- If a single comment contains multiple Points, assign separate labels.
- If a reviewer raises new Points later, give them new labels.

2.  For each author reply, assign the label(s) of the Point(s) it responds
to.
- If one reply addresses multiple Points, list all labels separated by
commas (e.g., "Label 2, 5").
- 'Strength' typically does not require a reply.  Focus on matching replies
to 'weaknesses' and 'questions'.

# OUTPUT FORMAT
Produce exactly {num} lines, one per part, in order:
Part k:  Label X
or, if multiple:
Part k:  Label X, Y

{Segmented reviewer comments and author responses}
```

## Prompts for Review Classification (Part 1)

```
You are given a reviewer comment (and optionally the authors' response).
Your task is to classify the comment using the following taxonomy.  The
output should only contain the sub-category name.
{Segmented Individual Item Review Comments}
# Taxonomy
1.  QUAL (Quality):  Is the submission technically sound?  Are claims well
supported (e.g., by theoretical analysis or experimental results)?  Are
the methods used appropriate?  Is this a complete piece of work or work
in progress?  Are the authors careful and honest about evaluating both the
strengths and weaknesses of their work?
- QUAL-MET (Methodological Soundness):  Evaluates whether the proposed
algorithms, models or system architectures are technically correct and free
of conceptual or implementation errors.
- Strength:  Highlights that the method is well-formulated, mathematically
consistent and correctly implemented; no major flaws are apparent.
- Weakness:  Points out conceptual errors, mis-specified objectives,
algorithmic flaws or misuses of optimisation methods.
- QUAL-EXP (Experimental Design & Evaluation):  Assesses the adequacy of
the experimental setup:  choice of datasets, baselines, evaluation metrics,
ablation studies, and statistical tests.
- Strength:  Notes comprehensive experiments, appropriate baselines,
sufficiently large data and proper metrics; experiments convincingly
validate claims.
- Weakness:  Points out inadequate baselines, missing standard benchmarks,
inappropriate metrics or lack of ablation studies.
- QUAL-REP (Reproducibility & Implementation Details):  Considers whether
the submission provides enough detail (e.g., hyperparameters, code
availability, dataset splits) to replicate the work and whether the
implementation follows best practices.
- Strength:  Commends open-sourced code, detailed hyperparameters, clear
dataset descriptions and comprehensive training details enabling easy
reproduction.
- Weakness:  Notes lack of code, incomplete hyperparameters, unspecified
data splits or other missing details that hinder reproduction.
- QUAL-CMP (Comparisons to Prior Work):  Evaluates whether the paper
sufficiently compares against relevant prior work and state-of-the-art
methods.
- Strength:  Acknowledges thorough comparisons against recent and
appropriate baselines and proper citation of related work.
- Weakness:  Notes missing comparisons to well-known recent methods,
outdated baselines, or incomplete literature review.
- QUAL-STA (Statistical Rigor & Validation):  Evaluates whether statistical
analyses (e.g., significance tests, confidence intervals) are properly
applied and whether reported improvements are statistically meaningful.
- Strength:  Praises appropriate statistical tests, robust validation,
confidence intervals and transparent reporting of variance.
- Weakness:  Criticises absence of statistical tests, reliance on single
runs, or misuse of statistical methods.
```

Prompts for Review Classification (Part 2)

```
2.  CLAR (Clarity):  Is the submission clearly written?  Is it well
organized?  (If not, please make constructive suggestions for improving
its clarity.)  Does it adequately inform the reader?  (Note that a superbly
written paper provides enough information for an expert reader to reproduce
its results.)
- CLAR-WRT (Writing, Terminology & Algorithm Presentation):  Covers overall
 writing quality and organization, precision of terminology and key concepts,
 and the clarity of algorithm presentations (code snippets, pseudocode,
 workflow diagrams).
- Strength:  Well-structured sections and smooth narrative; concise,
 readable prose; all specialized terms and assumptions are clearly defined
 and consistently used; pseudocode/code/workflows are self-contained,
 well-documented (clear variable names, inputs/outputs), and easy to follow
 step-by-step.
- Weakness:  Unclear or disorganized exposition, excessive jargon or
 redundancy; missing/ambiguous definitions or misuse of terms; confusing or
 incomplete pseudocode/code/workflows (undefined parameters, missing steps,
 opaque variable names).
- CLAR-NOT (Notation & Mathematical Explanation Clarity):  Considers whether
 mathematical notation is consistent, well defined and explained.
- Strength:  Highlights clear definitions, consistent notation and
 well-explained derivations.
- Weakness:  Notes inconsistent symbols, missing definitions or unexplained
 mathematical steps.
- CLAR-FIG (Figures & Visual Aids Clarity):  Evaluates whether figures,
 tables, plots and visualisations are legible, properly labelled, and aid
 understanding.
- Strength:  Praises informative and well-designed figures that clearly
 illustrate results or architectures.
- Weakness:  Points out illegible plots, missing labels, confusing colour
 schemes or misleading visualisations.
```

---

**Prompts for Review Classification (Part 3)**

3. SIGN (Significance): Are the results impactful for the community? Are
others (researchers or practitioners) likely to use the ideas or build on
them? Does the submission address a difficult task in a better way than
previous work? Does it advance our understanding/knowledge on the topic in
a demonstrable way? Does it provide unique data, unique conclusions about
existing data, or a unique theoretical or experimental approach?
– SIGN-BRD (Broad Research Impact): Evaluates whether the work addresses a
broadly important problem or advances understanding in a way that is likely
to influence multiple research areas. – Strength: Notes that the work
tackles a significant, widely relevant challenge or introduces a paradigm
that may influence many researchers.
– Weakness: Suggests that the work addresses a niche or already
well-explored problem with limited broader interest.
– SIGN-DOM (Domain/Applied Impact): Assesses the significance of the
contribution for a specific application domain (e.g., NLP, robotics,
healthcare).
– Strength: Highlights how the method outperforms existing approaches or
addresses an important challenge in a specialised field.
– Weakness: Notes limited domain relevance, inadequate demonstration on
domain-specific benchmarks or unconvincing domain benefits.
– SIGN-SOT (Improvement Over State of the Art): Focuses on the magnitude
and importance of improvements relative to current state-of-the-art methods.
– Strength: Praises substantial, consistent performance gains over strong
baselines with appropriate statistical support.
– Weakness: Criticises marginal or statistically insignificant improvements,
or improvements only on favourable datasets.
– SIGN-IMP (Real-World & Societal Impact): Considers the potential
practical and societal ramifications of the work, including benefits, risks,
fairness, environmental impacts and accessibility.
– Strength: Recognises thoughtful discussion of societal benefits, harms,
fairness, environmental impact and possible mitigations.
– Weakness: Points out a lack of consideration of societal impact or
failure to discuss who benefits and who may be harmed.

4. ORIG (Originality): Does the work provide new insights, deepen
understanding, or highlight important properties of existing methods? Is
it clear how this work differs from previous contributions, with relevant
citations provided? Does the work introduce novel tasks or methods that
advance the field? Does this work offer a novel combination of existing
techniques, and is the reasoning behind this combination well-articulated?
As the questions above indicates, originality does not necessarily
require introducing an entirely new method. Rather, a work that provides
novel insights by evaluating existing methods, or demonstrates improved
efficiency, fairness, etc. is also equally valuable.

---

**Prompts for Review Classification (Part 4)**

```
 - ORIG-PROB (Novel Problem Formulation):  Comments on the introduction of a
new problem, task, or dataset.  Includes innovative problem definitions that
highlight previously unaddressed challenges.
- Strength:  Highlights creative and meaningful new problem formulations or
datasets that open new avenues for research.
- Weakness:  Notes that the problem is a minor variation of existing tasks
or lacks clear motivation.
- ORIG-MTH (Novel Methodology or Algorithm):  Assesses whether the paper
introduces a genuinely new algorithmic approach or architectural design
rather than a minor tweak of existing methods.
- Strength:  Recognises innovative algorithmic designs with clear conceptual
advances.
- Weakness:  Points out that the method is a straightforward modification or
re-implementation of existing techniques.
- ORIG-ANL (Novel Analysis or Insights):  Pertains to original theoretical
insights or analyses that deepen understanding of existing methods or
phenomena.
- Strength:  Notes insightful analysis that sheds new light on known methods
or uncovers unexpected behaviours.
- Weakness:  Suggests that the analysis is trivial, already known, or does
not yield meaningful insights.
- ORIG-EXP (Novel Experimental Setup or Data):  Evaluates whether the paper
proposes new experimental setups, benchmarks, evaluation protocols or
collects new datasets.
- Strength:  Praises introduction of meaningful new datasets, benchmarks or
evaluation protocols.
- Weakness:  Notes that the experimental setup is derivative or does not add
value beyond existing benchmarks.
- ORIG-COM (Creative Combination of Existing Methods):  Considers whether
the paper combines well-known techniques in an original way and whether the
rationale behind the combination is compelling.
- Strength:  Recognises innovative synergies or architectures that combine
methods in a way that yields new capabilities.
- Weakness:  Suggests that the combination is superficial or lacks a clear
justification for being novel.
- ORIG-NEG (Negative Results or Critical Assessments):  Covers papers
that provide critical evaluations, ablations or negative results showing
limitations of existing methods.
- Strength:  Commends thoughtful critical analysis or the presentation of
negative results that challenge assumptions.
- Weakness:  Notes superficial or poorly justified criticism, or negative
results that lack rigour.
```

**Prompts for Review Classification (Part 5)**

```
5.  POL (Policy/Compliance):  Encompasses policy- or compliance-related
concerns such as ethics, data/privacy compliance, anonymity rules,
plagiarism, licensing, and broader impact.  These issues often require
checking adherence to conference or legal policies and may involve ethical
considerations.
- POL-ETH (Ethics & Responsible AI Compliance):  Addresses ethical concerns,
 including fairness, bias, harms to marginalised populations, dual use and
 whether the authors appropriately discuss and mitigate ethical risks.
- Strength:  Notes thorough consideration of potential harms, fairness,
 privacy and mitigation strategies.
- Weakness:  Flags unaddressed ethical issues, potential harms or
 insufficient discussion of biases.
- POL-DAT (Data Usage & Privacy Compliance):  Considers whether the data
 used in the paper comply with privacy regulations and licensing terms (e.g.,
 consent, personally identifiable information).
- Strength:  Praises adherence to data-use policies, proper anonymisation
 and appropriate licensing.
- Weakness:  Points out potential privacy violations, insufficient data
 consent or misuse of licensed data.
- POL-ANO (Double-Blind or Anonymity Violations):  Concerns whether the
 submission inadvertently reveals author identities or institutional
 affiliations, violating double-blind review policies.
- Strength:  Confirms that the paper appropriately anonymises authors and
 affiliations.
- Weakness:  Highlights self-identifying text, acknowledgements or URLs that
 compromise anonymity.
- POL-PLG (Plagiarism & Dual Submission):  Addresses issues of plagiarism,
 self-plagiarism, or dual submission to multiple venues.
- Strength:  Confirms originality of content, proper citations and no
 evidence of duplicate submission.
- Weakness:  Notes textual or figure overlap, uncredited reuse of material
 or simultaneous submission.
- POL-IMP (Broader Impact & Societal Considerations):  Evaluates whether the
 authors complete required checklists and discuss the broader impact of their
 work on society.
- Strength:  Comprehensive and honest broader-impact statements and required
 checklists; documented IRB/ethics approvals where applicable; explicit
 and correct software/data licences (and usage within licence terms); clear
 consent/approval identifiers and compliance evidence.
- Weakness:  Missing or boilerplate impact statements; absent/unclear
 required checklists; missing IRB or ethics approval where needed; ambiguous
 or incompatible licensing; evidence of non-compliant use (e.g., violating
 dataset or code licence terms).
```

## Prompts for Review Classification (Part 6)

```
### Boundary Rules for Disambiguation
1.  CLAR-WRT vs.  QUAL-EXP: Comments about missing baselines or inadequate
experiments belong under QUAL-EXP, even when poor writing makes experiments
hard to follow; issues solely about readability go under CLAR-WRT.
2.  CLAR-FIG vs.  QUAL-EXP: Critiques about illegible or unclear plots go
to CLAR-FIG; critiques about missing plots or missing baseline curves go to
QUAL-EXP.
3.  QUAL-REP vs.  CLAR-NOT: Missing code, hyperparameters or data splits
fall under QUAL-REP; undefined variables or inconsistent notation fall under
CLAR-NOT.
4.  SIGN-IMP vs.  POL-IMP: General discussion of societal benefits or harms
without mandatory checklists goes to SIGN-IMP; comments about compliance
with required impact statements go to POL-IMP.
5.  ORIG-EXP vs.  QUAL-EXP: Introducing a new dataset or benchmark is
ORIG-EXP; critiquing the thoroughness of experiments on existing datasets
is QUAL-EXP.
6.  POL-ETH vs.  SIGN-IMP: Ethical violations, bias or harm to marginalised
groups belong in POL-ETH; high-level impact comments without compliance
issues belong in SIGN-IMP.
7.  QUAL-CMP vs.  SIGN-SOT: Missing baselines or incomplete literature
reviews are QUAL-CMP; judging whether reported improvements are meaningful
goes to SIGN-SOT.
8.  ORIG-COM vs.  ORIG-MTH: Combining existing methods in an original way is
ORIG-COM; proposing entirely new algorithms is ORIG-MTH.
9.  QUAL-CMP vs.  QUAL-EXP: Missing/irrelevant or outdated baselines and
literature mapping → QUAL-CMP; evaluation protocol design (metrics, splits,
tuning fairness, test coverage) → QUAL-EXP.
10.  ORIG-MTH vs.  QUAL-EXP: Whether the algorithm/architecture is truly
novel vs.  a minor tweak → ORIG-MTH; whether experiments sufficiently
validate the (novel or not) method → QUAL-EXP.

---

**Output:** Return **only** the applicable label(s).  If the reviewer
mentions multiple distinct comments, assign different labels, separated
by commas.  If the reviewer mentions only one comment, assign the most
reasonable single label.  Do not include any other words or explanation in
the output.
```

**Prompts for Secondary Segmentation**

```
You are given a list of {sen_num} sentences from a peer-review discussion,
each prefixed with "ID: {{id}}".

TASK:
Classify each sentence into one or more of the following candidate labels:
{f"Label: {lbl}, Explanation: {label_explanations[lbl]}" for lbl in
candidate_labels}

GUIDELINES:
- A sentence can have multiple labels if it discusses multiple aspects
simultaneously
- Use only the provided candidate labels; do not invent new ones
- For all sentences within the same opinion, assign exactly the same label
or set of labels to every sentence in that opinion
- Be precise and concise in your judgment

OUTPUT FORMAT:
For each of the {sen_num} sentences, output exactly one line:
Sentence k:  Label X
or if multiple labels apply:
Sentence k:  Label X, Y

Process exactly {sen_num} sentences in the order provided.
```

Prompts for Review Clustering

```
You are a professional academic peer review analysis assistant.
Your ONLY task is to assign discussion point IDs based on the *identical
specific subject*
being discussed in each opinion.  Extract maximally granular topics (e.g.,
'Random Forest algorithm',
'SGD learning rate value').  CRITICAL: Opinions with contradictory stances
on the EXACT SAME subject.
MUST share the same ID. Different subjects receive different IDs, regardless
of opinion similarity.
Focus exclusively on subject identity, not reviewer names or opinion
agreement."
"Please assign discussion point IDs to reviewer opinions based on *identical
discussion content*.
Task Requirements:
1.  Review ALL reviewer opinions across ALL blocks below
2.  Extract the *most specific core subject* from each opinion (e.g.,
'Random Forest algorithm selection', 'SGD learning rate value')
3.  Assign an ID (starting from 1) to each *unique specific subject*
4.  **CRITICAL**:  Reuse the same ID for opinions discussing the *exact same
subject*, even if their evaluations are opposite/contradictory
5.  Different specific subjects receive different IDs, even if semantically
related
Content Identity Rules:
- **Subject must match precisely**: 'SGD is suitable' vs.  'SGD is
unsuitable' = SAME ID (identical subject:  SGD suitability)
- **Maximal specificity required**:  Use 'F1-score metric' or 'L2
regularization strength', NOT broad terms like 'methodology' or 'evaluation'
- **Different subjects, different IDs**:  Comments on 'data augmentation' vs.
'train-test split' = DIFFERENT IDs
- ID assignment depends *solely* on the identity of the discussed subject,
completely independent of opinion stance or reviewer name
Output Format (JSON):
- Return a mapping of block_id -> reviewer_name -> point_id
- Only include blocks that have reviewer opinions
Example:
```json
{
"blocks":  [
{
"block_id":  0,
"reviewer_assignments":  {
"Reviewer 1":  1,
}
}
]
}
```
Text Blocks to Analyze:
```

## Prompts for Inter-Reviewer Conflict Identification

```
You are a professional academic peer review analysis assistant.
Focus ONLY on detecting conflicts between reviewer opinions within each
discussion point.
Be precise about whether viewpoints are truly contradictory vs.
complementary.

Please analyze reviewer opinions within each discussion point to identify
GENUINE conflicts.
### Rules for Conflict Detection:
1.  **Definition of Conflict:**
- Direct contradiction on the IDENTICAL aspect (e.g., Reviewer A says
'method is novel', Reviewer B says 'method lacks novelty').
- Opposing sentiments (Positive vs. Negative) regarding the same specific
feature.

2.  **What is NOT a Conflict (Crucial):**
- **Granularity Differences:** Specific examples (e.g., 'works on
large images') vs. General statements (e.g., 'generalizes well') are
COMPLEMENTARY, not conflicting.
- **Different Aspects:** Opinions on different dimensions (e.g.,
'efficiency' vs 'accuracy') are not conflicts.
- **Non-Argumentative Text:** Ignore conflicts involving purely factual
lists, citations, references, or formatting artifacts (e.g., 'NO.', block
IDs).
- **Same Reviewer:** Opinions from the same reviewer name typically refine
or list details for their own points and should not be flagged as conflicts.

Task:
For each discussion point ID, examine all associated reviewer opinions.
Step 1:  Filter out non-opinion text (citations, meta-data).
Step 2:  Check if opinions express opposing views on the same core issue.
Step 3:  Output conflict status.

Output Format (JSON):
- point_conflicts:  point_id:  "has_conflict":  bool

Discussion Point Groups:
{prompt_parts}
```

Prompts for Inter-Reviewer Conflict Resolution

```
You are a professional academic peer review analysis assistant.
Focus on resolving conflicts between opinions. Use ONLY block_id for
identification.
Respond with ONLY the required JSON format.

You are an expert academic peer review analysis assistant. Your task is to
resolve conflicts between reviewer opinions on a specific discussion point.
### Conflict Resolution Guidelines:
1. **Identify the Core Issue**: Determine the exact technical/academic
point of disagreement.
2. **Evaluate Each Position**: Assess the validity of each opinion based
on logical consistency, technical soundness, and alignment with academic
standards.
3. **Make a Judgment**: Determine which opinion block(s) have the more
correct/valid position.

**Important:** Use block_id (e.g., 'block_5') to identify opinions in your
response.

=== META-REVIEW (Overall Context) ===
{meta review} --- Discussion Point point_id ---
{reviewer points}
--- Output Format (JSON ONLY) ---
{
"correct_blocks": ["block_X", ...],
"incorrect_blocks": ["block_Y", ...]
}
```

Prompts for Reviewer-Author Conflict Identification

```
You are a professional academic peer review analysis assistant.
Focus on detecting if authors explicitly refute reviewer opinions.
Analyze carefully but only output the JSON array as requested.

You are analyzing a peer review conversation.
f"Reviewer '{reviewer_name}' has {len(opinions)} opinions in this paper."
Your task is to determine if the author's responses explicitly refute each
opinion.

### CRITICAL GUIDELINES FOR 'REFUTATION':
1.  **What is a Refutation?**
- The author explicitly argues that the reviewer's comment is **factually
incorrect**, **irrelevant**, or based on a **misunderstanding**.
- Signals:  'We disagree', 'The reviewer is incorrect', 'This is a
misunderstanding', 'We do not think this is necessary'.

2.  **What is NOT a Refutation (False Positives to AVOID):**
- **Acceptance + Differentiation:** If the reviewer asks to compare with a
baseline (e.g., ConViT [6]), and the author adds the comparison but explains
*why* their method is different/better (e.g., 'Unlike ConViT, we do X...'),
this is **COMPLIANCE**, not refutation.
- **Technical Clarification:** Phrases like 'However, our method...'  used
to explain technical boundaries vs.  baselines are NOT refutations of the
reviewer.
- **Citation/Metadata:** If the 'opinion' text is merely a reference string
(e.g., '- [6] d'Ascoli...'), it is data, not an argument.  It CANNOT be
refuted.  **Always return False for citations.**

Task:
Check if the author explicitly rejects the validity of the reviewer's
specific point.  If they accepted the task (even with caveats) or if the
text is just a citation, return False.

=== META-REVIEW (for overall context) ===
{meta_review}
=== AUTHOR'S RESPONSES (All combined) ===
{author_response}

=== REVIEWER OPINIONS TO ANALYZE ===
{reviewer_opinion}
=== OUTPUT FORMAT ===
Return a JSON array where each element corresponds to each opinion above:
[
{"opinion_index":  0, "refutes":  true},
{"opinion_index":  1, "refutes":  false}
...
]

Only output the JSON array, no explanations.
```

## Prompts for Reviewer-Author Conflict Resolution

```
You are a precise academic validator.  Focus on factual accuracy and logical
soundness.
Your judgment should be based on evidence in the text, not assumptions.
Output strictly in JSON format with a 'judgments' array containing one entry
per opinion.

You are an expert academic peer review validator.
Your task is to determine which of the reviewer's criticisms are
**factually/logically incorrect** based on the author's response and
meta-review.

### CONTEXT:
f"**Meta-Review:** {meta_review if meta_review.strip() else '(No meta-review
available)'}"
"", f"**Author's Response (the refutation):** author_response"
"", f"**Reviewer's Opinions (being refuted):** opinions_str"

### INSTRUCTIONS:
1.  **Reviewer is WRONG (true)** if:
- The reviewer's claim is based on factual errors about the paper
- The reviewer misunderstood the methodology/results
- The reviewer's request is logically inconsistent or impossible
- Meta-review explicitly or implicitly supports the author's position

2.  **Reviewer is NOT wrong (false)** if:
- The reviewer's concern is valid but author disagrees on priority/scope
- The author is making excuses without addressing the core issue
- The meta-review sides with the reviewer or remains neutral
- It's a matter of interpretation rather than factual error

### OUTPUT FORMAT:
Return ONLY a JSON object with a 'judgments' array:
{
"judgments":  [
{
"block_id":  "block_id",
"is_reviewer_wrong":  true/false,
},
...
]
}
```

Prompts for ReviewSplit Model

```
You are given a peer-review discussion split into {num} parts, each prefixed
with "Part {{id}}:".  Each part is one comment in the discussion.

# GOAL
Assign numeric labels to EVERY part so that all parts about the same
discussion Point share the same label.

Definitions and rules:
- A Point is one atomic strength, weakness, or question raised by a reviewer
(or discussed in replies).
- Labels are integers starting from 1 ("Label 1", "Label 2", ...).
- If a reviewer raises new Points later in the discussion, assign new labels
to those Points.  Try to avoid using the same label for Points that are very
far apart in the discussion, even if they are related.
- For summary-only parts that are neither strengths nor weaknesses, treat
the entire summary as one single Point and assign one label to the whole
summary.

# OUTPUT FORMAT
Produce exactly {num} lines, one per part, in input order:
Part k:  Label X

{Segmented reviewer comments}
```

Prompts for ReviewClassify Model

```
You are given a reviewer comment (and optionally the authors' response).
Your task is to classify the comment using the following taxonomy.  The
output should only contain the sub-category name.
# Taxonomy
...  ...
### Boundary Rules for Disambiguation
1.  CLAR-WRT vs.  QUAL-EXP: Comments about missing baselines or inadequate
experiments belong under QUAL-EXP, even when poor writing makes experiments
hard to follow; issues solely about readability go under CLAR-WRT.
2.  CLAR-FIG vs.  QUAL-EXP: Critiques about illegible or unclear plots go
to CLAR-FIG; critiques about missing plots or missing baseline curves go to
QUAL-EXP.
3.  QUAL-REP vs.  CLAR-NOT: Missing code, hyperparameters or data splits
fall under QUAL-REP; undefined variables or inconsistent notation fall under
CLAR-NOT.
4.  SIGN-IMP vs.  POL-IMP: General discussion of societal benefits or harms
without mandatory checklists goes to SIGN-IMP; comments about compliance
with required impact statements go to POL-IMP.
5.  ORIG-EXP vs.  QUAL-EXP: Introducing a new dataset or benchmark is
ORIG-EXP; critiquing the thoroughness of experiments on existing datasets
is QUAL-EXP.
6.  POL-ETH vs.  SIGN-IMP: Ethical violations, bias or harm to marginalised
groups belong in POL-ETH; high-level impact comments without compliance
issues belong in SIGN-IMP.
7.  QUAL-CMP vs.  SIGN-SOT: Missing baselines or incomplete literature
reviews are QUAL-CMP; judging whether reported improvements are meaningful
goes to SIGN-SOT.
8.  ORIG-COM vs.  ORIG-MTH: Combining existing methods in an original way is
ORIG-COM; proposing entirely new algorithms is ORIG-MTH.
9.  QUAL-CMP vs.  QUAL-EXP: Missing/irrelevant or outdated baselines and
literature mapping → QUAL-CMP; evaluation protocol design (metrics, splits,
tuning fairness, test coverage) → QUAL-EXP.
10.  ORIG-MTH vs.  QUAL-EXP: Whether the algorithm/architecture is truly
novel vs.  a minor tweak → ORIG-MTH; whether experiments sufficiently
validate the (novel or not) method → QUAL-EXP.

---

**Output:** Return **only** the applicable label(s).  If the reviewer
mentions multiple distinct comments, assign different labels, separated
by commas.  If the reviewer mentions only one comment, assign the most
reasonable single label.  Do not include any other words or explanation in
the output.

Reviewer Comment:
{reviewer_comments}
```

## E.2. Prompts for CoCoreviewBench Evaluation

We present all the prompts used in our paper in §5 in the following tables, including:

- **Prompts for Paper-Level and Category-Level Metrics of CoCoReviewBench with LLM-as-a-Judge**: We input an AI review together with the corresponding human references. This prompt is used to evaluate the quality of the input AI review by comparing it against the human references. The prompt design largely follows Garg et al. (2025) and Sadallah et al. (2025). In addition, as mentioned in the main text, Correctness and Thoroughness are scored by comparison with human references to reduce LLM-as-a-Judge bias. To ensure the judge does not rely on its own knowledge but instead strictly compares with humans, we make slight wording changes: we rename Correctness to Alignment, so the LLM focuses on consistency with humans, and rename Thoroughness to Completeness, so the LLM focuses on breadth and depth relative to humans, enabling a more robust evaluation.

- **Prompts for General AI Reviewer**: For proprietary AI reviewers, we directly use the prompts specified in their papers/code. For general models, we follow Lu et al. (2024) and use their prompt to generate the final review.

- **Prompts for Our Trained AI Reviewer Agent**: For our trained AI reviewer agent, we design prompts to be as simple as possible, changing only the input taxonomy to switch among review agents for different categories.

Prompts for Paper-Level and Category-Level Metric of CoCoreviewBench with LLM-as-a-Judge (Part 1)

```
You are an expert evaluator.
Given the gold reference comment and a candidate comment, score the
candidate's actionability, grounding_specificity, verifiability, alignment,
completeness and clarity on a scale of 1-5 (or X where specified).
You will perform the evaluation in two steps:
(1) First extract whether the candidate comment is a claim or a normal
statement (no claim).
(2) Then evaluate the candidate on the specified aspects using the provided
1{5 (or X) scales, following the flow rule.

### **Step 1:  Claim Extraction**

**Objective:**
Determine whether the given text contains a claim (i.e., an opinion,
judgment, or suggestion) or consists solely of factual statements that
require no verification.

**Claim Definition:**
A statement is considered a claim if it falls into one or more of the
following categories:
- **Subjective opinions or disagreements** (e.g., criticism of an
experimental choice).
- **Suggestions or requests for changes** (e.g., recommending removal,
addition, or discussion).
- **Judgments about the paper** (e.g., stating something is unclear, not
well-written, or lacks detail).
- **Deductions or inferred observations** that go beyond merely stating
facts.
- **Statements requiring justification** to be understood or accepted.

**Normal Statements ("No Claim")**
A statement is considered a normal statements if it falls into one or more
of the following categories:
- Describes facts without suggesting changes.
- Makes general statements about the paper without an opinion.
- Presents objective, verifiable facts that require no justification.
- Asks for clarifications or general questions.
- States logical statements or directly inferable information.
- Makes positive claims (e.g., *\The paper is well-written"*), as these do
not help improve the work.

**Flow Rule**
If the candidate comment is a Normal Statement, then:
- actionability = "X"
- verifiability = "X"
- Still evaluate the other aspects (grounding_specificity, alignment,
completeness, clarity).
Otherwise (it contains a claim), evaluate all aspects normally.
```

Prompts for Paper-Level and Category-Level Metric of CoCoreviewBench with LLM-as-a-Judge (Part 2)

```
### **Step 2: Evaluation**

**Objective:**
Evaluate the candidate comment across multiple quality dimensions
(actionability, grounding_specificity, verifiability, alignment,
completeness, and clarity) using the provided definitions and 1{5 (or X)
scales.
If Step 1 classifies the candidate as a **Normal Statement ("No Claim")**,
skip actionability and verifiability and directly set both to "X".

Aspect: actionability
**Actionability**

**Definition:** Measures the level of actionability in a review point.  We
evaluate actionability based on two criteria:

1.  **Explicit vs.  Implicit**:
- **Explicit:** Actions or suggestions that are direct or apparent.  Authors
can directly identify modifications they should apply to their draft.
Clarification questions should be treated as explicit statements if they
give a direct action.
- **Implicit:** Actions that need to be inferred from the comment.  This
includes missing parts that need to be added.  Authors can deduce what needs
to be done after reading the comment.

2.  **Concrete vs.  Vague**:
- **Concrete:** Once the action is identified, the authors know exactly what
needs to be done and how to apply the action.
- **Vague:** After identifying the action, the authors still don't know how
to carry out this action.

**Importance:** It's more important for actions to be concrete so that
authors know how to apply them.  It's also preferred for actions to be
stated directly rather than inferred.
```

Prompts for Paper-Level and Category-Level Metric of CoCoreviewBench with LLM-as-a-Judge (Part 3)

```
 **Actionability Scale (1-5 & X):**

1.  **1:  Unactionable**
- **Definition:** The comment lacks meaningful information to help authors
improve the paper.  Authors do not know what they should do after reading
the comment.

2.  **2:  Borderline Actionable**
- **Definition:** The comment includes an implicitly stated action or an
action that can be inferred.  However, the action itself is vague and lacks
detail on how to apply it.

3.  **3:  Somewhat Actionable**
- **Definition:** The comment explicitly states an action but is vague on
how to execute it.

4.  **4:  Mostly Actionable**
- **Definition:** The comment implicitly states an action but concretely
states how to implement the inferred action.

5.  **5:  Highly Actionable**
- **Definition:** The comment contains an explicit action and concrete
details on how to implement it.  Authors know exactly how to apply it.

X. **X: No claim**
- **Definition:** The comment contains only factual, descriptive statements
without claims, opinions, or suggestions.

Aspect:  grounding_specificity
**Grounding Specificity**

**Definition:** Measures how explicitly a review comment refers to a
specific part of the paper and how clearly it identifies the issue with that
part.  This helps authors understand what needs revision and why.  Grounding
specificity has two key components:
```

Prompts for Paper-Level and Category-Level Metric of CoCoreviewBench with LLM-as-a-Judge (Part 4)

1. **Grounding:** How well the authors can identify the specific part of the paper being addressed.
– **Weak Grounding:** The author can make an educated guess but cannot precisely identify the referenced part.
– **Full Grounding:** The author can accurately pinpoint the section, table, figure, or unique aspect being addressed. This can be achieved through:
– Literal mentions of sections, tables, figures, etc.
– Mentions of unique elements of the paper.
– General comments that clearly imply the relevant parts without explicitly naming them.

2. **Specificity:** How clearly the comment details what is wrong or missing in the referenced part. If external work is mentioned, it also evaluates whether specific examples are provided.

**Importance:** It's more important for the comment to be grounded than to be specific.

**Grounding Specificity Scale (1–5):**

1. **Not Grounded**
– **Definition**: This comment is not grounded at all. It does not identify a specific area in the paper. The comment is highly unspecific.

2. **Weakly Grounded and Not Specific**
– **Definition**: The authors cannot confidently determine which part the comment addresses. Further, the comment does not specify what needs to be addressed in this part.

3. **Weakly Grounded and Specific**
– **Definition**: The authors cannot confidently determine which part the comment addresses. However, the comment clearly specifies what needs to be addressed in this part.

4. **Fully Grounded and Under-Specific**
– **Definition**: The comment explicitly mentions which part of the paper it addresses, or it should be obvious to the authors. However, this comment does not specify what needs to be addressed in this part.

5. **Fully Grounded and Specific**
– **Definition**: The comment explicitly mentions which part of the paper it addresses, and it is obvious to the authors. The comment specifies what needs to be addressed in this part.

## Prompts for Paper-Level and Category-Level Metric of CoCoreviewBench with LLM-as-a-Judge (Part 5)

```
Aspect: verifiability
**Verifiability**

**Definition:** Assess how well a claim is verified by examining the
reasoning, common knowledge, or external references provided. The purpose
is to ensure that the review comment helps the authors improve their work.

**Verification Methods:**
A claim is considered verifiable if supported by one or more of the
following:
- **Logical reasoning** { A clear explanation of why the claim is valid.
- **Common knowledge** { Reference to well-accepted practices or standards.
- **External references** { Citation of relevant literature, data, or
sources.

**Verifiability Scale (1{5 & X):**

1. **1: Unverifiable**
- The comment contains a claim without any supporting evidence or
justification.
2. **2: Borderline Verifiable**
- Some support is provided, but it is vague, insufficient, or difficult to
follow.
3. **3: Somewhat Verifiable**
- The claim has some justification but lacks key elements (e.g., examples,
references).
4. **4: Mostly Verifiable**
- The claim is well-supported but has minor gaps in explanation or
references.
5. **5: Fully Verifiable**
- The claim is thoroughly supported by explicit, sufficient, and robust
evidence, such as:
- Clear reasoning and precise explanations.
- Specific references to external works.
- Logical and unassailable common-sense arguments.
X. **X: No Claim**
- The comment contains only factual, descriptive statements without claims,
opinions, or suggestions.
```

Prompts for Paper-Level and Category-Level Metric of CoCoreviewBench with LLM-as-a-Judge (Part 6)

```
Aspect:  alignment
**Alignment**

**Definition:** Measures how well a candidate review comment aligns with
the gold reference opinions in terms of *directional stance* (agreement
vs disagreement) and *whether it expresses the same underlying opinion*.
Alignment is not about wording similarity; it is about whether the candidate
supports the same idea/critique/praise as the gold reference.

**Alignment Scale (1{5):**

1.  **1:  Opposing Opinion**
- **Definition:** The candidate expresses an opposing stance to the key
gold opinion(s) (i.e., contradicts the main thrust), or argues the opposite
conclusion on the same issue.

2.  **2:  Mostly Misaligned**
- **Definition:** The candidate is largely misaligned:  it either
contradicts some key gold opinion(s) or frames the issue in a way that
conflicts with the gold.

3.  **3:  Mixed or Unrelated**
- **Definition:** The candidate includes a mixture of aligned and opposing
opinions, **or** it is largely unrelated/orthogonal (neither clearly aligned
nor clearly opposing).  It does not consistently support the gold's stance.

4.  **4:  Mostly Aligned**
- **Definition:** The candidate is mostly aligned with the gold's stance and
opinions, but may miss some constraints/details, soften/shift emphasis, or
contain minor deviations that do not overturn the main agreement.

5.  **5:  Fully Aligned**
- **Definition:** The candidate expresses the same underlying opinion(s) as
the gold reference:  it matches the main concerns/praises in both direction
and substance, without material contradictions.
```

---

Prompts for Paper-Level and Category-Level Metric of CoCoreviewBench with LLM-as-a-Judge (Part 7)

```
Aspect:  completeness
**Completeness**
**Definition:** Measures how completely a candidate review comment covers
the gold reference opinions (human reviewer comments).  Completeness is
about *coverage breadth* across the set of gold opinions:  how many distinct
gold concerns/praises are meaningfully addressed by the candidate.

Completeness has two components:

1.  **Coverage Breadth:**
- Whether the candidate addresses few vs many of the key gold opinions.

2.  **Coverage Adequacy:**
- Whether the candidate meaningfully addresses the covered gold opinions
 (not just a vague mention).
- If the candidate is much more generic than a gold opinion, that gold
 opinion should be treated as *not covered* or only weakly covered.

**Importance:** Breadth is more important than length.  A long candidate
can still be incomplete if it misses key gold opinions.

**Completeness Scale (1{5):**

1.  **1:  Minimally Complete**
- **Definition:** The candidate covers none or almost none of the key gold
 opinions.  Most gold opinions are not covered; any overlap is accidental or
 extremely weak.

2.  **2:  Low Completeness**
- **Definition:** The candidate covers only a small portion of gold
 opinions, and/or covers them weakly (generic mentions without addressing
 the underlying request).  Several key gold opinions are missing.

3.  **3:  Moderate Completeness**
- **Definition:** The candidate covers some gold opinions, but misses
 important ones.  Coverage may be uneven:  some opinions covered, others only
 weakly covered.

4.  **4:  High Completeness**
- **Definition:** The candidate covers most key gold opinions, with
 generally adequate substance.  Only a small number of less-central gold
 opinions are missing or weakly covered.

5.  **5:  Fully Complete**
- **Definition:** The candidate covers nearly all key gold opinions, and
 coverage is meaningful (not merely topical).  Few or no gold opinions remain
 not covered.
```

Prompts for Paper-Level and Category-Level Metric of CoCoreviewBench with LLM-as-a-Judge (Part 8)

```
Aspect: clarity
**Clarity**

**Definition:** Judges how coherent, readable, and well-structured the
comment is. Clarity focuses on delivery: logical flow, ease of reading,
and whether the language is concise rather than confusing or verbose.

**Clarity Scale (1{5):**

1. **1: Unclear / Hard to Read**
- **Definition:** The comment is confusing or difficult to follow due to
poor coherence (disorganized, fragmented, or contradictory), heavy verbosity
with little signal, or unclear phrasing. Readers cannot easily identify the
main point.

2. **2: Mostly Unclear**
- **Definition:** The comment has substantial readability issues (wordy,
tangled sentences, weak structure). The main point is present but requires
effort to extract, and the delivery obscures meaning.

3. **3: Moderately Clear**
- **Definition:** The comment is generally readable and the main point can
be understood, but delivery is imperfect (some verbosity, uneven structure,
minor coherence gaps). It is understandable but not crisp.

4. **4: Clear**
- **Definition:** The comment is coherent and easy to read, with a mostly
well-structured presentation. Minor verbosity or small structural issues
may exist, but they do not hinder comprehension.

5. **5: Very Clear / Concise**
- **Definition:** The comment is concise, well-structured, and highly
readable. Ideas flow logically, wording is efficient, and the main point
(and any request) is immediately understandable.

###Instruction:
Be more objective and conservative in your grading.
Respond strictly as JSON, do not provide any other content:
{
"claim_extraction": "claim" or "normal",
"actionability": <int 1-5> or "X",
"grounding_specificity": <int 1-5>,
"verifiability": <int 1-5> or "X",
"alignment": <int 1-5>,
"completeness": <int 1-5>,
"clarity": <int 1-5>
}
```

---

### Prompts for General AI Reviewer (Part 1)

You are an AI researcher who is reviewing a paper that was submitted to a
prestigious ML venue.
Be critical and cautious in your decision.

## Review Form
Below is a description of the questions you will be asked on the review form
for each paper and some guidelines on what to consider when answering these
questions.
When writing your review, please keep in mind that after decisions have been
made, reviews and meta-reviews of accepted papers and opted-in rejected
papers will be made public.

1.  Summary:  Briefly summarize the paper and its contributions.  This is
not the place to critique the paper; the authors should generally agree with
a well-written summary.
– Strengths and Weaknesses:  Please provide a thorough assessment of the
strengths and weaknesses of the paper, touching on each of the following
dimensions:
– Originality:  Are the tasks or methods new?  Is the work a novel
combination of well-known techniques?  (This can be valuable!)  Is it
clear how this work differs from previous contributions?  Is related work
adequately cited
– Quality:  Is the submission technically sound?  Are claims well supported
(e.g., by theoretical analysis or experimental results)?  Are the methods
used appropriate?  Is this a complete piece of work or work in progress?
Are the authors careful and honest about evaluating both the strengths and
weaknesses of their work
– Clarity:  Is the submission clearly written?  Is it well organized?  (If
not, please make constructive suggestions for improving its clarity.)  Does
it adequately inform the reader?  (Note that a superbly written paper
provides enough information for an expert reader to reproduce its results.)
– Significance:  Are the results important?  Are others (researchers
or practitioners) likely to use the ideas or build on them?  Does the
submission address a difficult task in a better way than previous work?
Does it advance the state of the art in a demonstrable way?  Does it provide
unique data, unique conclusions about existing data, or a unique theoretical
or experimental approach?

2.  Questions:  Please list up and carefully describe any questions and
suggestions for the authors.  Think of the things where a response from the
author can change your opinion, clarify a confusion or address a limitation.
This can be very important for a productive rebuttal and discussion phase
with the authors.

Prompts for General AI Reviewer (Part 2)

```
3. Limitations: Have the authors adequately addressed the limitations and
potential negative societal impact of their work?  If not, please include
constructive suggestions for improvement.
In general, authors should be rewarded rather than punished for being
up front about the limitations of their work and any potential negative
societal impact.  You are encouraged to think through whether any critical
points are missing and provide these as feedback for the authors.

4. Ethical concerns:  If there are ethical issues with this paper,
please flag the paper for an ethics review.  For guidance on when this is
appropriate, please review the NeurIPS ethics guidelines.

5. Soundness:  Please assign the paper a numerical rating on the following
scale to indicate the s oundness of the technical claims, experimental and
research methodology and on whether the central claims of the paper are
adequately supported with evidence.
4:  excellent
3:  good
2:  fair
1:  poor

6. Presentation:  Please assign the paper a numerical rating on the
following scale to indicate the quality of the presentation.  This
should take into account the writing style and clarity, as well as
contextualization relative to prior work.
4:  excellent
3:  good
2:  fair
1:  poor

7. Contribution:  Please assign the paper a numerical rating on the
following scale to indicate the quality of the overall contribution this
paper makes to the research area being studied.  Are the questions being
asked important?  Does the paper bring a significant originality of ideas
and/or execution?  Are the results valuable to share with the broader
NeurIPS community.
4:  excellent
3:  good
2:  fair
1:  poor

8. Overall:  Please provide an "overall score" for this submission.
Choices:
```

Prompts for General AI Reviewer (Part 3)

```
 10:  Award quality:  Technically flawless paper with groundbreaking
impact on one or more areas of AI, with exceptionally strong evaluation,
reproducibility, and resources, and no unaddressed ethical considerations.
9:  Very Strong Accept:  Technically flawless paper with groundbreaking
impact on at least one area of AI and excellent impact on multiple areas
of AI, with flawless evaluation, resources, and reproducibility, and no
unaddressed ethical considerations.
8:  Strong Accept:  Technically strong paper with, with novel ideas,
excellent impact on at least one area of AI or high-to-excellent impact
on multiple areas of AI, with excellent evaluation, resources, and
reproducibility, and no unaddressed ethical considerations.
7:  Accept:  Technically solid paper, with high impact on at least one
sub-area of AI or moderate-to-high impact on more than one area of AI, with
good-to-excellent evaluation, resources, reproducibility, and no unaddressed
ethical considerations.
6:  Weak Accept:  Technically solid, moderate-to-high impact paper, with
no major concerns with respect to evaluation, resources, reproducibility,
ethical considerations.
5:  Borderline accept:  Technically solid paper where reasons to accept
outweigh reasons to reject, e.g., limited evaluation.  Please use sparingly.
4:  Borderline reject:  Technically solid paper where reasons to reject,
e.g., limited evaluation, outweigh reasons to accept, e.g., good evaluation.
Please use sparingly.
3:  Reject:  For instance, a paper with technical flaws, weak evaluation,
inadequate reproducibility and incompletely addressed ethical
considerations.
2:  Strong Reject:  For instance, a paper with major technical flaws, and/or
poor evaluation, limited impact, poor reproducibility and mostly unaddressed
ethical considerations.
1:  Very Strong Reject:  For instance, a paper with trivial results or
unaddressed ethical considerations

9.  Confidence:  Please provide a "confidence score" for your assessment
of this submission to indicate how confident you are in your evaluation.
Choices:
5:  You are absolutely certain about your assessment.  You are very familiar
with the related work and checked the math/other details carefully.
4:  You are confident in your assessment, but not absolutely certain.  It is
unlikely, but not impossible, that you did not understand some parts of the
submission or that you are unfamiliar with some pieces of related work.
3:  You are fairly confident in your assessment.  It is possible that you
did not understand some parts of the submission or that you are unfamiliar
with some pieces of related work.  Math/other details were not carefully
checked.
2:  You are willing to defend your assessment, but it is quite likely that
you did not understand the central parts of the submission or that you are
unfamiliar with some pieces of related work.  Math/other details were not
carefully checked.
1:  Your assessment is an educated guess.  The submission is not in your
area or the submission was difficult to understand.  Math/other details were
not carefully checked.
```

## Prompts for General AI Reviewer (Part 4)

```
Below are some sample reviews, copied from previous machine learning
conferences.
Note that while each review is formatted differently according to each
reviewer's style, the reviews are well-structured and therefore easy to
navigate.
Paper:

```
{paper_text}
```

Review:

```
{review_text}
```
Here is the paper you are asked to review:
```
{paper_text}
```
```

Prompts for Our Trained AI Reviewer Agent

```
You are an AI researcher who is reviewing a paper that was submitted to a
prestigious ML venue.  Be critical and cautious in your decision.  Focus
your review strictly on the following dimension:
{taxonomy[k]}

Here is the paper you are asked to review:
```
{paper_text}
```

## Review Form

Below is a description of the specific dimension you are asked to evaluate
for the paper.
When writing your review, please keep in mind that after decisions have been
made, reviews and meta-reviews of accepted papers and opted-in rejected
papers will be made public.

- Assessment Dimension:  Please provide a thorough assessment of the
 strengths and weaknesses of the paper, focusing strictly on the following
 dimension:
{taxonomy[k]}

- Questions:  Please list up and carefully describe any questions and
 suggestions for the authors **related to the dimension above**.  Think of
 the things where a response from the author can change your opinion, clarify
 a confusion or address a limitation regarding this specific aspect.
```

