# OpenReview forum: "CoCoReviewBench: A Completeness- and Correctness-Oriented Benchmark for AI Reviewers"
_ICML.cc/2026/Conference — ICML 2026 regular_

### Official Review · Reviewer_1oVe · 2026-02-16

**Soundness:** 2
**Presentation:** 3
**Significance:** 2
**Originality:** 3
**Overall Recommendation:** 3
**Confidence:** 4

**Summary:**

This paper presents CoCoReviewBench, a new benchmark aimed at more reliably assessing AI-based paper reviewers. The paper pinpoints a core weakness in current evaluation setups: they typically treat human reviews as gold-standard references, even though these reviews are frequently incomplete and can contain factual inaccuracies or misunderstandings. To mitigate this, the paper designs a detailed taxonomy with 23 subcategories and scores AI reviews only on the specific aspects that the human review actually covers, thereby avoiding unfair penalties. To tackle the problem of correctness, the paper cleverly exploits the natural peer-review process—focusing on disagreements between reviewers and on author rebuttals, as ultimately resolved by meta-reviews—to filter out erroneous human judgments. Based on a curated corpus of 3,900 NeurIPS and ICLR papers, the study shows that current AI reviewers fall short of humans in both accuracy and thoroughness, while also underscoring the unique strengths of reasoning-oriented models for peer-review applications.

**Compliance With Llm Reviewing Policy:**

Affirmed.

**Final Justification:**

My concerns have been adequately addressed.

**Key Questions For Authors:**

1. I did not find a GitHub repository link or any code and data availability statement in the submission. Was omitting the anonymous link an oversight, or is there another reason the code is not provided?
2. Relying on LLMs to build a benchmark that is then used to assess other LLMs raises concerns about circular dependence and potential bias. While you did include human verification, manually reviewing only 50 out of 3,900 papers seems insufficient to ensure the quality of the full dataset. Have you considered this scaling limitation, and how do you resolve the potential noise in the unverified data?
3. The paper defines a highly granular 23-category taxonomy tailored specifically for ML conference submissions. Does this specific taxonomy and your conflict-resolution pipeline generalize well to other scientific domains or distinct computer science subfields (such as databases or systems)? Clarifying the framework's extensibility beyond the core ML domain would greatly enhance the paper's broader impact on the community.
4. Because the dataset relies heavily on public OpenReview data (NeurIPS 2021-2024, ICLR 2017-2025), there is a severe risk of data contamination, as the evaluated LLMs likely memorized these reviews during pre-training or the RLHF phase. How do you control for this to ensure the models are genuinely reasoning rather than just recalling text?

**Limitations:**

Yes

**Strengths And Weaknesses:**

Strengths:
S1: The authors establish a highly motivated, necessary, and practical research direction for the reliable assessment of LLMs in academic settings.
S2: The construction of a two-level taxonomy with 23 fine-grained subcategories enables a much fairer, category-specific evaluation.
S3: The paper provides a thorough and insightful empirical analysis across various open-source and closed-source LLMs. It yields valuable insight for the community, such as the superiority of reasoning models in verifiability, and the surprising hallucination risks (e.g., critiquing figures) in text-only AI reviewers.
Weakness:
W1: The authors use meta-reviews to resolve conflicts between reviewers and to determine the correct ground truth. However, in real-world peer review, ACs typically concentrate on high-level decisions (accept/reject) and do not systematically resolve every detailed, atomic technical disagreement. As a result, treating the meta-review as an absolute ground truth may incorrectly validate specific points, thereby introducing bias.
W2: The paper enhances correctness by discarding reviews deemed unreliable due to conflicts. However, a high-quality AI reviewer could offer deep, out-of-distribution insights that human reviewers fail to notice. Within the current setup, such novel insights might be penalized or ignored because they lack a corresponding human reference, thereby limiting the evaluation of an AI's true potential to outperform humans.
W3: The dataset consists of papers from NeurIPS (2021–2024) and ICLR (2017–2025). However, the evaluated models (e.g., GPT-5, Gemini-3, Llama-3) have likely seen these publicly available OpenReview discussions during pre-training. As a result, the LLMs may be recalling existing reviews instead of producing them independently. The fairness of this comparison has to be discussed.
W4: The paper defines 23 fine-grained subcategories. While highly detailed, individual review sentences often span several of these subcategories (for instance, highlighting an experimental flaw that also implicates the methodology). Such granularity increases classification confusion. Moreover, Figure 5 uses complex stacked and sorted bar charts, and Figure 7 contains dense multi-panel charts. With 5 major categories, 23 subcategories, and densely packed, tilted X-axis labels, it is challenging for readers to quickly identify which specific areas the performs poorly.

---

> ### Author Rebuttal · Authors · 2026-03-31
>
> > W1: Using meta-reviews to resolve conflicts may introduce bias because they operate at a high level.
>
> See our response to Reviewer MTgQ W2/Q1.
>
> > W2: Novel AI insights beyond human references may be unfairly penalized under the current benchmark.
>
> Thank you for raising this issue. This problem is widespread in prior benchmarks. Our paper does not introduce this problem. Rather, we aim to alleviate it by making fuller use of existing human annotations.
>
> Specifically, we classify human references and evaluate at the category level, so that when human reviews lack comments in a category, we skip evaluation there, reducing penalization due to missing human comments. Besides, our evaluation dimensions include *Thoroughness*, which measures how comprehensively a comment discusses the current category. When an AI reviewer provides deeper discussion, it can receive a higher score on this dimension, reducing the penalty caused by missing reference content. Finally, our reference is an aggregated and deeply cleaned set of all human reviewer opinions, which further raises the upper bound of the opinions that can be meaningfully evaluated.
>
> > W3, Q4: Public OpenReview data may create contamination, making it unclear whether models are reasoning or recalling.
>
> Thank you for pointing out this issue. This is a common challenge in benchmark construction. For any benchmark built from public data, the evaluated models can in principle access that data, and once the dataset is released, future leakage is also difficult to avoid. This issue is not unique to our work. However, compared with prior work, our framework has one advantage: we have an automatic data construction pipeline that can build a dataset from the latest available data, on which the evaluated models have not yet been trained.
>
> > W4: The 23-way taxonomy may increase classification confusion, and the figures are hard to read.
>
> Thank you for pointing this out. We also observed the issue of classification confusion. To address it, we introduce the *Secondary Segmentation* step, which further splits existing segments that contain multiple categories. For sentences that still cannot be cleanly split, we allow them to belong to multiple categories and include them in all relevant references, thereby avoiding artificially low scores due to missing reference content. Our taxonomy construction process is also carefully designed to ensure classification quality; please see FzRF W4/Q1/Q5 for details.
>
> Regarding Figures 5 and 7, we apologize for the inconvenience. In the revised version, we will present only top-level category results in the figures and move the full 23-subcategory results into new tables.
>
> > Q1: The submission appears to be missing an anonymous code/data availability link.
>
> Thank you for pointing this out. Because the model artifacts and the full dataset are large and cannot be fully hosted on GitHub, we have released the code and part of the data at: https://anonymous.4open.science/r/CoCoReviewBench/. The remaining parts will be fully released upon acceptance.
>
> > Q2: Using LLMs in benchmark construction may introduce circular dependence, and validating only 50 papers may be insufficient.
>
> Thank you for raising this issue. We agree with this point, and throughout the generation of the reference, we avoid using LLMs to generate the reference itself. We only use them for classification and filtering, while the reference is still entirely generated by human reviewers.
>
> In addition, reviewing itself is also difficult and expensive for humans. Therefore, we sampled 50 papers, covering reviews from 180 reviewers and 1,700 opinions, to achieve as broad coverage as possible. Human verification on a randomly sampled subset is also common practice in large-scale benchmark construction [1,2]. We use such a subset to assess the reliability of the overall benchmark. Because the subset is selected at random, it is expected to be broadly representative of the full dataset, so the quality estimated on the sample can serve as a reasonable estimate of the quality of the full benchmark.
>
> [1] *MAVIS: Mathematical Visual Instruction Tuning with an Automatic Data Engine*. ICLR 2025.
>
> [2] *CoreEval: Automatically Building Contamination-Resilient Datasets with Real-World Knowledge toward Reliable LLM Evaluation*. ACL 2025.
>
> > Q3: It is unclear how well the taxonomy and conflict-resolution pipeline generalize beyond ML/AI peer review.
>
> Thank you for raising this question. Since our evaluation set is based on NeurIPS and ICLR, the current 23-category taxonomy is designed around the ML/AI reviewing setting and derived from ML/AI reviewer guidelines, so it is not expected to naturally fit other fields as-is. However, we do provide a procedural guideline for constructing such a taxonomy, which can in principle be used to derive a high-quality taxonomy for arbitrary domains. We hope this alleviates your concern. Please see our response to Reviewer FzRF W4/Q1/Q5 for more details.

---

> > ### Author Rebuttal · Reviewer_1oVe · 2026-04-03
> >
> > I appreciate the authors' comprehensive rebuttal. I'd like to raise my score.

---

### Official Review · Reviewer_23pk · 2026-03-09

**Soundness:** 3
**Presentation:** 4
**Significance:** 3
**Originality:** 3
**Overall Recommendation:** 5
**Confidence:** 4

**Summary:**

This paper introduces CoCoReviewBench for more reliable evaluation of AI reviewers. It builds a two-level taxonomy with 23 subcategories spanning quality, clarity, significance, originality, and policy, segments human reviews into atomic opinions, and organizes them by category. It further leverages reviewer–author–meta-review discussions to identify disagreements and adjudications, filtering out unreliable opinions to strengthen reference correctness. The benchmark curates 3,900 papers from ICLR and NeurIPS, and trains lightweight models to segment and classify AI reviews, enabling category-level, multi-dimensional, fine-grained evaluation and analysis.

**Compliance With Llm Reviewing Policy:**

Affirmed.

**Final Justification:**

The authors' rebuttal has successfully addressed my primary concerns. Therefore, I maintain my original score of 5.

**Key Questions For Authors:**

1. Coverage limitation:Reference correctness relies primarily on “explicit conflicts + meta-review adjudication,” but many erroneous opinions may never surface as conflicts (e.g., missing author responses, overly mild rebuttals, or points not addressed by other reviewers). As a result, a non-trivial number of undetected erroneous references may remain, making the boundary of the claim that the references are “overall more correct” less clear.

2. Reliability of the adjudication signal: Meta-reviews are often brief and may not adjudicate specific disputed points or provide supporting evidence. Moreover, the authors’ human verification indicates lower accuracy for detecting reviewer–author conflicts on rejected papers, which may weaken the credibility of treating meta-reviews as expert annotations.

3. End-to-end robustness: Although step-level agreement and human spot-checks are reported, the multi-stage LLM pipeline (segmentation → clustering → classification → conflict detection → adjudication) is prone to cascading errors. The lack of an end-to-end sensitivity/robustness analysis (e.g., swapping the model in one stage or perturbing key thresholds and observing changes in the final references and evaluation rankings) leaves the stability of the conclusions insufficiently supported.

**Limitations:**

yes

**Strengths And Weaknesses:**

**Soundness**

Strengths: From a soundness perspective, the paper presents a fairly systematic pipeline for improving the completeness and correctness of human-review references. It includes a two-level taxonomy with 23 subcategories, atomic-opinion segmentation with secondary verification, conflict resolution based on reviewer/author/meta-review disagreements, and validation via cross-model agreement and human spot-checking. The experiments further provide category-level, multi-dimensional evaluation across multiple types of AI reviewer models. Overall, the claims are supported by a coherent experimental setup, and the methodological choices align well with the stated goals.

Weaknesses: One caveat is that the correctness enhancement relies primarily on “explicit conflicts + meta-review adjudication” as expert signals. While the paper acknowledges coverage and reliability boundaries in its limitations, it may still miss unexpressed errors that never surface as explicit disagreements and therefore cannot be filtered. Overall, the work is methodologically and empirically solid, and it is relatively cautious in describing its limitations and scope of applicability.

**Presentation**

Strengths: The paper is generally clearly written and well structured. It first motivates the problem by highlighting the incompleteness and inconsistency of using human reviews as gold references, then presents the design principles, construction pipeline, and verification procedures of CoCoReviewBench, and finally supports its claims with experiments and fine-grained analyses. The narrative is easy to follow.

Weaknesses: While the Related Work provides a fairly comprehensive review of AI reviewer corpora and evaluation methods, it does not clearly position the proposed benchmark relative to closely related benchmarks/structured datasets of a similar type. I suggest adding a comparison table against the most relevant existing resources/benchmarks (e.g., whether they include atomic-opinion segmentation, category taxonomy labels, use of rebuttal/meta-review, conflict adjudication and erroneous-opinion annotations, etc.), and explicitly clarifying the key novel contributions and the non-substitutable aspects of CoCoReviewBench compared to prior resources.

**Significance**

Strengths：The paper focuses on an important and practical problem: how to reliably evaluate AI reviewers. By introducing a structured taxonomy and reference cleaning based on discussion-level disagreements, it provides a finer-grained and actionable evaluation framework. This can improve the community’s ability to understand and measure AI review quality (e.g., correctness, thoroughness, and grounding) and lays a foundation for building more trustworthy training signals and future agentic reviewer systems.

Weaknesses: End-to-end error propagation in the multi-step LLM pipeline is not quantified.
Although the authors report step-level agreement and human spot-checks, the overall pipeline involves multiple stages—segmentation → clustering → classification → conflict detection → adjudication. Errors at any stage can cascade and potentially alter the final references and evaluation results. The lack of a systematic end-to-end robustness/sensitivity analysis (e.g., swapping the model used in one stage, or perturbing key thresholds) makes it difficult to assess the stability of the conclusions.

**Originality**

The paper offers a fairly clear new perspective on AI reviewer evaluation and benchmark construction. It decomposes the shortcomings of using human reviews as gold references into two actionable issues—incompleteness and incorrectness—and accordingly proposes category-level evaluation together with a reference-cleaning mechanism based on disagreements in reviewer–author–meta-review discussions to improve reference correctness. Meanwhile, the paper constructs a structured benchmark that includes atomic-opinion segmentation, fine-grained category labels, similar-opinion clustering, and erroneous-opinion annotations, covering a large-scale collection of papers from ICLR and NeurIPS. If the data and annotations are publicly released, it would provide valuable practical support to the field at the levels of evaluation and data resources. Overall, the contribution is better viewed as an innovative combination of existing ideas rather than a single-point algorithmic breakthrough, and the construction pipeline and motivating logic are relatively coherent.

---

> ### Author Rebuttal · Authors · 2026-03-31
>
> > Soundness Weakness, Q1: Conflict-based filtering may miss many unexpressed errors.
>
> Thank you for recognizing that the work is methodologically and empirically solid, and cautious in scope of applicability. We also mentioned this weakness you provided in the limitations. When constructing the benchmark, we also considered this issue and selected papers for which at least 75% of the opinions received replies, so as to ensure that at least most opinions have author responses as annotation signals. Furthermore, we do not claim that we filter out all errors. Rather, we make as full use as possible of existing human annotation data to provide a higher-quality reference, although there may still be imperfections. For a more detailed discussion, please see our reply to Reviewer MTgQ W2/Q1.
>
> > Presentation Weakness: The benchmark should be positioned more clearly against related resources, ideally with a comparison table and a clearer novelty statement.
>
> Thank you for this very concrete and helpful suggestion. It helps us clarify our originality, and the result is shown in the following table. On the correctness side, to our knowledge, no prior AI reviewer benchmark has addressed conflict adjudication and incorrect-opinion annotations together. Some prior work includes meta-reviews and author rebuttals in the benchmark, but does not use them to detect incorrect opinions. On the completeness side, some work constructs categorical metrics for category-level evaluation, but to our knowledge no prior work has explicitly identified the completeness problem of the reference, and no prior work uses a taxonomy as fine-grained as our 23 subcategories. These differences show that our benchmark substantially advances prior approaches and provides a meaningful improvement over existing resources.
>
> |Resource|Atomic opinions|Fine-grained labels|Category-level evaluation|Rebuttal signal|Meta signal|Conflict detection|Incorrect annotation|
> |---|---|---|---|---|---|---|---|
> |PeerRead[1]||~||||||
> |MReD[2]||✓|||✓|||
> |NLPeer[3]||||||||
> |SEA[4]||||||||
> |ReviewMT[5]||||✓|✓|||
> |Re²[6]||||✓|✓|||
> |Aspects[7]|✓|✓||||||
> |RottenReviews[8]||✓||||||
> |GRE-bench[9]||||||||
> |RevUtil[10]|✓|✓|✓|||||
> |CoCoReviewBench(ours)|✓|✓|✓|✓|✓|✓|✓|
>
> [1] *A Dataset of Peer Reviews (PeerRead): Collection, Insights and NLP Applications*. NAACL 2018.
>
> [2] *MReD: A Meta-Review Dataset for Structure-Controllable Text Generation*. Findings of ACL 2022.
>
> [3] *NLPeer: A Unified Resource for the Computational Study of Peer Review*. ACL 2023.
>
> [4] *Automated Peer Reviewing in Paper SEA: Standardization, Evaluation, and Analysis*. Findings of EMNLP 2024.
>
> [5] *Peer Review as A Multi-Turn and Long-Context Dialogue with Role-Based Interactions*. arXiv preprint, 2024.
>
> [6] *Re²: A Consistency-ensured Dataset for Full-stage Peer Review and Multi-turn Rebuttal Discussions*. arXiv preprint, 2025.
>
> [7] *Identifying Aspects in Peer Reviews*. Findings of EMNLP 2025.
>
> [8] *RottenReviews: Benchmarking Review Quality with Human and LLM-Based Judgments*. CIKM 2025.
>
> [9] *Benchmarking LLMs' Judgments with No Gold Standard*. ICLR 2025.
>
> [10] *The Good, the Bad and the Constructive: Automatically Measuring Peer Review’s Utility for Authors*. EMNLP 2025.
>
> > Q2: The adjudication signal may be unreliable because meta-reviews are brief and human verification is weaker on rejected papers.
>
> Thank you for pointing this out. For the discussion that meta-reviews are often high-level summaries, please see our response to Reviewer MTgQ W2/Q1. Regarding the lower agreement on this step, we conducted a careful manual inspection and provide case studies in Appendix C.2. We find that these cases typically involve a negative reviewer comment for which the author has already given a fairly convincing rebuttal. Accordingly, our annotation pipeline treats the reviewer opinion as potentially problematic and removes it. However, human annotators tend to place more weight on the overall negative tone of the meta-review and therefore judge the reviewer’s negative opinion as reasonable. In our view, under such cases, removing the disputed opinion in our automatic pipeline is itself reasonable, which suggests that the true accuracy may be higher than the reported value.
>
> > Significance Weakness, Q3: The multi-stage pipeline needs stronger end-to-end robustness analysis.
>
> Thank you for recognizing our step-level evaluation experiments. Furthermore, our human annotation are in fact obtained based on the end-to-end annotation result of our final references. In this process, we consider all possible errors in our final reference, including classification, clustering, and conflict adjudication. These can reflect the correctness of each step, but in essence this is an analysis of our final benchmark. We hope this human annotation results can alleviate your concern and demonstrate the quality of our benchmark. Furthermore, regarding the error propagation issue, please see our response to Reviewer FzRF W1/Q2.

---

> > ### Author Rebuttal · Reviewer_23pk · 2026-04-01
> >
> > Thank you to the authors for the detailed response. I will maintain my original score: **Overall Recommendation: 5 (Accept)**.

---

### Official Review · Reviewer_FzRF · 2026-03-12

**Soundness:** 3
**Presentation:** 3
**Significance:** 3
**Originality:** 2
**Overall Recommendation:** 5
**Confidence:** 4

**Summary:**

This paper introduces a benchmark designed to evaluate AI-based peer review systems with a focus on both completeness and correctness of review feedback. The authors analyze human peer reviews from ICLR and NeurIPS and argue that human reviews are often incomplete and sometimes incorrect, making them unreliable as gold references for evaluating AI reviewers. To address this, the benchmark constructs a taxonomy of review aspects and decomposes reviews into atomic opinions that are categorized into 23 subcategories.
The benchmark also attempts to detect incorrect opinions by identifying conflicts between reviewers, authors, and meta-reviews, using the meta-review as the final adjudicator.  Using this benchmark, the authors evaluate several LLM-based reviewer systems and show that current AI reviewers still lag behind humans in correctness and thoroughness.

**Compliance With Llm Reviewing Policy:**

Affirmed.

**Final Justification:**

After reviewing the authors’ additional clarifications, I am now convinced that my main concerns have been sufficiently addressed, and I have accordingly increased my score.

In particular, the authors provided a much clearer and more comprehensive analysis of the pipeline validation. The additional breakdown of error propagation across different chains, along with quantitative evidence showing that upstream errors are not the dominant source of downstream failures, significantly strengthens my confidence in the robustness of the approach.
Furthermore, the new analysis of meta-review quality provides a more nuanced and convincing justification for using meta-reviews as an adjudication signal. While they are not perfect, the empirical evidence showing that, in strong-conflict cases, meta-reviews provide a meaningful and reasonably grounded signal (with most scoring ≥3 across multiple dimensions) supports their practical utility. I
Overall, the authors added meaningful new analysis that directly addresses the core of my concerns. This substantially improves my confidence in both the validity of the pipeline and the soundness of the design choices.

**Key Questions For Authors:**

How were the 23 subcategories defined and validated? Were human annotators involved in defining or verifying this taxonomy?

The benchmark relies heavily on splitting reviews into atomic opinions. Have you consider evaluating this step, and how the error in this step will be stacked into later steps of this process?
See "FENICE: Factuality Evaluation of summarization based on Natural language Inference and Claim Extraction" for evaluation of atomic fact extractions.

The framework assumes that meta-reviews provide the correct resolution when conflicts occur. If reviewers are prune to error, so does the meta reviewer. What evidence supports this assumption that meta reviewer is a better choice for ground truth, and how often do meta-reviews actually resolve the specific conflicts identified?

What is the distribution of accepted vs. rejected papers in the dataset? Could this distribution influence the types of review comments observed or the conclusions drawn about AI reviewers?

Prompts seems to be very detailed. How were the prompts for the LLM-based agents designed? Were they manually crafted, optimized with automated methods?

**Limitations:**

The authors provide a limitations section discussing the limitations of conflict-based filtering and the computational overhead of the evaluation pipeline. However, the paper could further discuss potential biases introduced by relying on meta-reviews as ground truth.

**Strengths And Weaknesses:**

**strengths**
- Evaluating AI reviewer systems is an increasingly important and timely topic as LLM-based reviewing tools are becoming more common in the scientific workflow.

- The benchmark provides a substantial dataset and the scale is solid for conducting research on this topic.

- The decomposition of reviews into atomic opinions provides more interpretable assessments of the reviews and I think it will be useful for analyzing reviewer behavior.

**Weaknesses:**
- Important steps such as atomic opinion splitting, classification accuracy, and conflict detection appear central to the benchmark construction but receive limited quantitative evaluation.

- The framework treats meta-reviews as the final arbiter in reviewer–author conflicts, which is questionable. Meta-reviews themselves may be subjective and inconsistent, and the paper does not sufficiently justify why they should be treated as reliable ground truth.

- Method description is not easy to follow. Several components (e.g., “prior injection via an agent framework” and the distillation of the agent workflow into an 8B model) are introduced without sufficient explanation, making it difficult to understand how the system operates.

- The paper introduces a taxonomy of 23 review subcategories, but it is not clearly explained how these categories were derived or validated.


Several closely related works on evaluating AI-generated peer reviews or AI-generated content in peer review are missing from the discussion, including:

- RottenReviews: Benchmarking Review Quality with Human and LLM-Based Judgments
- Is Your Paper Being Reviewed by an LLM? Benchmarking AI Text Detection in Peer Review
- MixRevDetect: Towards Detecting AI-Generated Content in Hybrid Peer Reviews
- Monitoring AI-Modified Content at Scale: A Case Study on the Impact of ChatGPT on AI Conference Peer Reviews

---

> ### Author Rebuttal · Authors · 2026-03-31
>
> > W1, Q2: The multi-step pipeline needs stronger quantitative validation, especially for atomic-opinion splitting and end-to-end robustness.
>
> Thank you for pointing this out. In Section 4.2.3, we report human verification results on 50 sampled papers, so atomic-opinion splitting, classification, and conflict detection have all been quantitatively evaluated. Appendix C.1 provides automatic verification for the key stages of the pipeline. We use 6 strong models on 180 held-out papers and perform leave-one-out cross-model validation. Their high agreement supports reliability.
>
> As for FENICE, its task differs from ours in two ways. First, we only segment the original review text, whereas FENICE involves rewriting and uses NLI to evaluate alignment between the opinion and the original text. Second, the granularity is different. We focus on atomic review claims, which may still contain sub-claims, whereas FENICE focuses on summarization, where finer-grained splitting is more natural. Therefore, we cannot directly use the human-annotated dataset used in FENICE for testing. To address this, we conduct human annotation and agreement evaluation above.
>
> Regarding error propagation, we design the pipeline to be conservative. It has three parts: segmenting and classifying opinions, detecting incorrect opinions through inter-reviewer conflicts, and detecting incorrect opinions through reviewer-author conflicts. The latter two depend on the first part, but not on each other, reducing the maximum depth. We also add a verification step (Secondary Split) to correct the under-segmentation failure in the first step. The human verification accuracy for this part is high, with 93.41% completely correct aggregation. Since later steps build on this high-accuracy step, serious error propagation is limited. In later steps, we first detect whether a conflict exists, and only then use the meta-review to decide whether an opinion is incorrect. In our preliminary experiments, this step-by-step execution made results reliable.
>
> > W2, Q3: Meta-reviews may be subjective and unreliable as the final signal for adjudicating reviewer-author conflicts.
>
> See our response to Reviewer MTgQ W2/Q1. In addition, we only use the meta-review when a conflict exists. In this case, the polarity of the meta-review for that category affects which side is finally selected. We always require the model to choose one side among opinions with inconsistent polarity, so conflicts almost never remain in the final reference.
>
> > W3: The method description is difficult to follow, especially the agent workflow and 8B distillation.
>
> We apologize for the confusion. “Prior injection via an agent framework” means that, instead of letting the model decide the workflow by itself, we design a more reliable workflow based on prior knowledge. “Distillation into an 8B model,” described in Section 4.3, refers to distilling the structured outputs generated by the earlier pipeline in Section 4.2 into an 8B model to reduce evaluation cost. We will further improve the presentation to make this clearer.
>
> > W4, Q1, Q5: The taxonomy and prompts are not clearly explained; their derivation and validation need to be clarified.
>
> Thank you for raising this issue. Details are provided in Appendix A.2. The 23 subcategories are not arbitrarily defined. We first refer to the NeurIPS Reviewer Guidelines and the ACL Rolling Review Review Form to determine the 5 top-level dimensions. We then use GPT-5 agent mode to extract subcategories from reviewer guidelines. We jointly optimize the taxonomy and the corresponding classification prompts on a small development set, using GPT-5 and Gemini-2.5-Pro and selecting the version with the higher inter-model agreement. After that, we further refine the prompts by adding boundary rules and again select the better version. In these processes, we generate multiple candidates, and the authors discuss them, select the best prompt, and perform category merging to avoid unclear boundaries. The final validation procedure is described in our response to W1/Q2 above.
>
> > W5: Related work on AI-generated or AI-modified peer review is missing from the discussion.
>
> Thank you for pointing this out. Most of them focus on detecting or analyzing AI-generated or AI-modified review text, providing a complementary perspective to our work. One paper also shows limited alignment between LLM-based and human evaluation, which focuses on evaluation dimensions rather than the references, and is complementary to ours. We will include a discussion of these works in the revised version.
>
> > Q4: The accepted/rejected distribution should be clarified, along with its possible impact on the conclusions.
>
> Thank you for raising this question. We have taken this issue into account. As described in Section 4.2.3, we linearly map scores to [0, 10], divide them into four segments using thresholds 3/5/7, and perform balanced sampling within each segment.

---

> > ### Author Rebuttal · Reviewer_FzRF · 2026-04-03
> >
> > Thank you for the detailed rebuttal and for addressing my questions. I appreciate the additional clarifications, especially regarding the validation of the multi-step pipeline and the explanation of the taxonomy construction and agent workflow.
> >
> > Overall, I believe several of my concerns have been partially addressed, but some aspects remain insufficiently resolved:
> >
> > - The added human verification and cross-model agreement are helpful and provide some evidence of reliability. However, the evaluation still feels somewhat limited in scale and depth, and it is not entirely clear how residual errors propagate through the pipeline or affect downstream conclusions. Given how central this pipeline is to the benchmark, stronger or more systematic validation would further strengthen confidence.
> > - The clarification that meta-reviews are only used when conflicts arise is helpful. However, the core concern remains that meta-reviews themselves can be subjective or incomplete, and the rebuttal does not fully justify why they should be considered a reliable adjudicator, especially in ambiguous cases.

---

> > > ### Author Response · Authors · 2026-04-08
> > >
> > > > Q1: Pipeline validation and error propagation
> > >
> > > Thank you for the question. Our human verification is not a set of isolated module validations, but an end-to-end validation of the final benchmark's correctness, which demonstrates the quality of our benchmark. To test whether residual errors propagate, we separate them into two chains: (1) classification -> reviewer-author conflicts (Cls->RA), and (2) classification -> grouping -> reviewer-reviewer conflicts (Cls->Grp->RR). For each chain, we compare downstream adjudication errors under clean versus imperfect upstream conditions.
> > >
> > > First, **the upstream structured steps are reliable**. On 50 human-annotated papers, review classification has 85.45% accuracy, and grouping has 93.41% accuracy.
> > >
> > > Second, **residual errors do propagate, but they are not the dominant source of downstream errors**. In the reviewer-author chain, the weighted adjudication error rate rises from 31.41% under clean upstream conditions to 47.45% under imperfect upstream conditions. In the reviewer-reviewer chain, the imperfect-upstream subset contains only 2 entries, so we do not report a separate weighted imperfect error rate. However, most downstream errors in both chains still occur under clean upstream conditions (>80%), as shown in the table below. Clean denotes the share of review words under clean upstream conditions, CleanErr and ImpErr denote downstream weighted error rates under clean and imperfect upstream conditions, and ErrFromUp denotes the proportion of downstream error words that arise from entries with upstream issues.
> > >
> > > |Chain|Clean|CleanErr|ImpErr|Δ|ErrFromUp|
> > > |---|---:|---:|---:|---:|---:|
> > > |Cls->RA|89.01%|31.41%|47.45%|+16.04%|15.73%|
> > > |Cls->Grp->RR|73.15%|24.63%|—|—|3.17%|
> > >
> > > Third, **the downstream impact is not uniformly distributed across papers**. Aggregating adjudication entries to the paper level, the average correctness ratio is 66.67%, but the distribution is uneven: 48.00% of papers are 100% correct, 18.00% are 0% correct, and only 12.00% fall in the middle 40%-60% range. This indicates that downstream errors are concentrated in a small subset of difficult papers rather than evenly affecting all papers. In turn, this suggests that the observed propagation penalty Δ should not be interpreted as arising solely from step-level error transmission: **part of the residual error arises from intrinsic task difficulty**, since papers with more ambiguous comments and discussion contexts are harder to adjudicate.
> > >
> > > Overall, residual errors do propagate, and this effect can be quantified on both correctness paths. However, given the relatively high accuracy of our early steps and the relatively limited strength of error propagation, propagation is not the only source, nor the dominant one.
> > >
> > > > Q2: The quality of meta-reviews as an adjudication signal
> > >
> > > Thank you for the question. We add a dedicated analysis of meta-review quality in conflict scenarios. We consider only papers with explicit conflict signals, and evaluate the original meta-review on four 1-5 dimensions, using LLM-as-a-Judge with the same settings as in Section 5. *Adjudication Clarity* measures whether the meta-review states a clear stance on the main disputed issue. *Groundedness* measures whether this stance is supported by reviewer comments and discussion context. *Verifiability* measures whether the judgment is concrete enough to be checked. *Conflict Coverage* measures whether the meta-review addresses the major disputed points rather than only part of them. A score of 5 denotes a highly verifiable meta-review that details all central disputes, whereas a score of 3 denotes a meta-review that already provides a meaningful overall signal about the conflict.
> > >
> > > We define strong-conflict papers as papers with at least five detected incorrect opinions. 73.1% of these have a four-dimensional average score of at least 3. This indicates that in most strongly conflicting papers, the meta-review provides a meaningful overall signal regarding the main dispute. The overall score is also higher for strong-conflict papers than for weak-conflict papers.
> > >
> > > |Conflict|Clarity|Ground|Verify|Cover|Avg.|
> > > |---|---:|---:|---:|---:|---:|
> > > |1-4|3.32|2.95|2.93|2.58|2.94|
> > > |>=5|3.60|3.27|3.23|2.85|3.24|
> > >
> > > These results suggest that meta-reviews can serve well as a high-level adjudication signal. However, we do not claim that meta-reviews are perfect. Their scores remain below the full-score standard, especially because Coverage is still relatively low. This suggests that meta-reviews are better understood as a high-level integration and adjudication signal, rather than fine-grained ground truth for every dispute.
> > >
> > > Therefore, our claim is not that meta-reviews are perfect, but that under the current peer-review process, if one wants the most reliable reference signal without introducing extra bias from LLM, there is in fact a lack of reliable alternatives, among which meta-reviews remain the strongest and most practical signals available.

---

### Official Review · Reviewer_MTgQ · 2026-03-13

**Soundness:** 3
**Presentation:** 3
**Significance:** 3
**Originality:** 3
**Overall Recommendation:** 4
**Confidence:** 4

**Summary:**

The paper studies the problem domain of designing an AI-based reviewer, a highly important question. The main premise, and departure from prior work, is that it questions ground truth human labels as gold standard, highlighting significant disagreement among human reviewers, imcompleteness and incorrectness. It resolves disagreements by using meta-reviews, and provides an LLM-as-a-Judge benchmark.

**Compliance With Llm Reviewing Policy:**

Affirmed.

**Final Justification:**

I thank the reviewers for their rebuttal and additional analysis. I am worried that the correlational analysis of completeness is confounded by other features, and that completeness itself does not capture meaningfully the quality of a peer review. While I am in favour of acceptance, I have my reservations, and keep my score.

**Key Questions For Authors:**

What is the quality of meta-reviews? As you treat them as ground truth, their correctness is important

**Limitations:**

yes

**Strengths And Weaknesses:**

Strengths:
 - Highly original question
 - Important question
 - Clear writing and exposition
 - Broad coverage of models

Weaknesses:
 1) Argument for the relevance of the incompleteness of human reviewers seem weak: Humans don't cover topics that they don't think are decision-relevant. This is rational and does not question the label of aggregated human judgment. Other evidence is needed to support this point.
 2) Using the meta-review, which is important for the method, as ground truth seems problematic as using human labels. For example, meta-reviews often don't cover a lot of topic areas either. Humans disagree, which is normal.
 3) The paper seems to contradict itself by saying "Our goal is not to synthesize reviews that are intrinsically better than humans." and "Green indicates better-than-human performance, while red indicates worse-than-human performance." (with the usual conventions of green being desirable and red not)

---

> ### Author Rebuttal · Authors · 2026-03-31
>
> > W1: It is unclear whether incompleteness is truly a problem, since reviewers may reasonably focus only on decision-relevant points.
>
> Thank you for pointing this out. We agree that, from the reviewer's perspective, covering only topics relevant to the decision is rational. However, when human reviews are used as the reference for evaluating AI reviews, the key question is whether valid comments are wrongly penalized because the reference is not comprehensive. An AI reviewer may provide comments that do not overlap with the human review but are still reasonable and useful, yet be treated as low-quality or irrelevant. This is why we argue that incompleteness can systematically affect evaluation. Our data support this point: a single reviewer covers only 5.10 out of 23 subcategories on average, and even after aggregating all reviewers, the average coverage is only 9.23 subcategories. This shows that human references, although sufficient for making a decision, are not complete. Therefore, our category-level evaluation is designed to reduce the risk of misinterpreting “not mentioned by humans” as “wrong if mentioned by AI.”
>
> > W2, Q1: Meta-reviews may be too coarse and subjective to serve as the adjudication signal for conflict resolution.
>
> Thank you for raising this critical issue. Our goal is to make the best possible use of existing human annotations, which are currently the most reliable and unbiased data source available to us, to construct a reference that is as reliable as possible. However, the resulting reference still cannot be considered perfect.
>
> Specifically, for LLM-as-a-judge evaluation to be reliable, it should rely on a reference to reduce bias. We cannot fully rely on LLM-generated content or judgments, since they may introduce bias. At the same time, directly using raw human reviews as the reference is also not sufficiently reliable because of limited completeness and correctness, yet this is the common setting in prior work. Based on the above analysis, we aim to use the human discussion process during review to construct a higher-quality reference without introducing LLM-generated opinions. However, the amount of available human annotation is still limited, making a perfect reference unrealistic at this stage. Therefore, our goal is not to create a perfect reference, but to make the best possible use of all available human annotations to improve its quality. Even after fully leveraging available human annotations, the reference we construct is still not perfect. Nevertheless, we believe that identifying this issue and providing a better evaluation scheme by making full use of existing data is itself an important contribution.
>
> Based on this observation, our work takes a step toward a more reliable reference using existing human annotations. In this process, although meta-reviews have limitations, as judgments written by more senior experts, they are still the strongest and most scalable human adjudication signal available in the current review process. Compared with individual reviewer comments, meta-reviews tend to have fewer issues in overall direction and correctness, although they are coarser and usually assess only higher-level dimensions. Therefore, our goal is not to ask the meta-review to provide full coverage, but to use its more accurate overall judgment about the category of the current conflict to select the side with the correct overall polarity, thereby filtering out incorrect opinions as much as possible. This use case does not require meta-reviews to fully adjudicate every detail; a high-level judgment is often sufficient.
>
> Therefore, the final result is a set of opinions whose level of detail is aligned with the aggregation of multiple human reviewers, while its high-level judgment is aligned with the meta-review, making it more correct and more comprehensive. Although meta-reviews may occasionally be wrong, they are already the most reliable signal available in peer review.
>
> > W3: The paper seems to blur the distinction between the goal of benchmark construction and above/below-human evaluation results.
>
> Thank you for pointing this out. These two statements refer to two different levels. The sentence “Our goal is not to synthesize reviews that are intrinsically better than humans” refers to the goal of benchmark construction. Our goal is not to synthesize a reference that is intrinsically better than humans, but to build a more reliable evaluation benchmark based on existing human review data. That is, we do not synthesize reviews, but only annotate/filter them. In contrast, “better-than-human performance / worse-than-human performance” in the experimental section refers to the evaluation result level, namely whether an AI review scores above or below the leave-one-out human baseline on a given metric. This is an objective empirical observation under our benchmark for evaluating AI reviewer systems, rather than the quality of the reference we construct.

---

> > ### Author Rebuttal · Reviewer_MTgQ · 2026-04-03
> >
> > Re W1: Your argument still sounds tautological to me -- you seem to be presenting evidence that completeness is important because there is no completeness. But that by itself is not an argument for completeness.
> > Re W2, Q1: This answer does not give me additional insight on how good metareviews are.
> > Re W3: Addressed, thank you.
> >
> > (Keeping my score)

---

> > > ### Author Response · Authors · 2026-04-08
> > >
> > > > W1: Incompleteness as an evaluation problem.
> > >
> > > Thank you for the question. We do not view incompleteness as a problem for human reviewing itself. For acceptance decisions, it is reasonable for reviewers to focus on issues they consider decision-relevant. The incompleteness problem we refer to is about evaluation: when human reviews are reused as references for evaluating AI reviewers, incompleteness can introduce systematic bias. A reasonable AI comment that is not mentioned by human reviewers can be judged as low-quality simply because the reference contains no matching signal.
> > >
> > > To test this directly, we ran an additional experiment under the same paper-level LLM-as-a-Judge protocol used in Section 5. For each paper, we split the generated AI review into two disjoint subsets based on the human reference categories: (1) category-consistent opinions, whose categories are covered by the human reference, and (2) category-inconsistent opinions, whose categories are absent from the human reference. We then evaluated these two subsets separately against the same paper-level human reference, keeping only papers where both subsets were non-empty and had comparable length (character ratio within 2x). Here, a negative delta means that the category-inconsistent subset receives a lower score than the category-consistent subset.
> > >
> > > |Model|ΔCorrectness|ΔThoroughness|ΔGrounding|ΔVerifiability|ΔClarity|ΔAvg.|
> > > |---|---:|---:|---:|---:|---:|---:|
> > > |Llama-3.1-8B-Instruct|-0.16|-0.27|-0.21|-0.07|-0.07|-0.15|
> > > |QwQ-32B|-0.37|-0.44|-0.23|-0.27|-0.19|-0.30|
> > > |DeepReviewer-7B|-0.16|-0.27|-0.09|-0.02|-0.09|-0.13|
> > > |Qwen3-32B|-0.37|-0.43|-0.37|-0.28|-0.24|-0.34|
> > > |Average|-0.27|-0.35|-0.23|-0.16|-0.15|-0.23|
> > >
> > > Across all four models, the category-inconsistent subset is scored systematically lower than the category-consistent subset. The largest drops appear in Correctness and Thoroughness. These dimensions depend directly on agreement with the reference, whereas the others mainly reflect general review quality. This suggests that the drop is not only due to potentially weaker review quality, but also reflects a systematic evaluation bias. Once human reviews are treated as the reference universe, comments from missing categories are systematically disadvantaged by paper-level evaluation.
> > >
> > > This is why the coverage statistics are relevant. Even after aggregation, human reviews cover only 9.23 out of 23 subcategories on average. This does not mean human reviewing itself is flawed. Rather, it shows that if evaluation treats human-mentioned content as a complete reference, the reference space is still highly sparse. Our category-level evaluation is designed to mitigate this mismatch.
> > >
> > > > W2: The quality of meta-reviews as an adjudication signal.
> > >
> > > Thank you for the question. We add a dedicated analysis of meta-review quality in conflict scenarios. We consider only papers with explicit conflict signals, and evaluate the original meta-review on four 1–5 dimensions, using LLM-as-a-Judge with the same settings in Section 5. *Adjudication Clarity* measures whether the meta-review states a clear stance on the main disputed issue. *Groundedness* measures whether this stance is supported by reviewer comments and discussion context. *Verifiability* measures whether the judgment is concrete enough to be checked. *Conflict Coverage* measures whether the meta-review addresses the major disputed points rather than only part of them. A score of 5 denotes a highly verifiable meta-review that details all central disputes, whereas a score of 3 denotes a meta-review that already provides a meaningful overall signal about the conflict.
> > >
> > > We define strong-conflict papers as papers with at least five detected incorrect opinions. 73.1% of these have a four-dimensional average score of at least 3. This indicates that in most strongly conflicting papers, the meta-review provides a meaningful overall signal regarding the main dispute. The overall score is also higher for strong-conflict papers than for weak-conflict papers.
> > >
> > > |Conflict|Clarity|Ground|Verify|Cover|Avg.|
> > > |---|---:|---:|---:|---:|---:|
> > > |1–4|3.32|2.95|2.93|2.58|2.94|
> > > |>=5|3.60|3.27|3.23|2.85|3.24|
> > >
> > > These results suggest that meta-reviews can serve reasonably well as a high-level adjudication signal. However, we do not claim that meta-reviews are perfect. Their scores remain below the full-score standard, especially because Conflict Coverage is still relatively low. This suggests that meta-reviews are better understood as a high-level integration and adjudication signal, rather than fine-grained ground truth for every detailed dispute.
> > >
> > > Therefore, our claim is not that human meta-reviews are perfect, but that under the current peer-review process, if one wants the most reliable reference signal possible without introducing extra bias from LLM-generated content, there is in fact a lack of reliable alternatives, among which meta-reviews remain the strongest and most practical human adjudication signals available.

---

### Decision · Program_Chairs · 2026-04-30

**Decision:**

Accept (regular)

**Comment:**

The paper designs a new benchmark for paper review quality, CoCoReviewBench. The authors argue that prior benchmarks incorrectly treat human reviews as ground truth, and penalize models for deviating from human reviews. Instead, the authors propose an elaborate automated processing pipeline, where the reviews are decomposed into atomic claims and opinions, that are cross-checked with other reviews, rebuttal, and the meta-review. One particular feature of the pipeline is the 23-category taxonomy: the authors tag the opinions from the reviews with one of 23 labels; then, the AI reviews can be evaluated on each category separately, without being penalized for raising points that are not covered by human reviews. With the benchmark, the authors evaluate multiple proprietary and open-weights models, and show that on Correctness and Thoroughness most models still underperform the human baseline. They also show that thinking models outperform non-thinking models, particularly on grounding and verifiability, and show that AI reviews are prone to hallucinations. To reduce the evaluation cost, they also distill their evaluation workflow into smaller 8B models.

Automated research review is both practically relevant, and an important capability for AI models to have. For example, it is required for enabling automated research and self-improvement loops. The authors design a highly thoughtful and elaborate pipeline for evaluating the AI review quality. The design decisions make sense and are well-supported by experiments. The empirical results on the quality of reviews from current models are interesting.

The reviewers raised several concerns, many of which were addressed in the rebuttal. For example, the authors added details on human verification for the quality of the benchmark. They also added results on the error sensitivity, and how upstream errors propagate downstream in the pipeline. Reviewers were also concerned that relying on meta-reviews may be problematic, and the authors clarified that meta-reviews are only used for conflict resolutions when there are contradictory opinions.

There are a few remaining concerns, that are harder to address. In particular, there is a contamination concern: the benchmark is based on public data, which was likely pretrained on, affecting conclusions about the model capabilities. Second, the size of the human study is limited (understandably, as it is labor-intensive).

Despite these limitations, the paper makes a meaningful contribution, and I think the proposed benchmark and the methodology for constructing the data will be useful to the ICML community.